# Learning the optimal Tikhonov regularizer for inverse problems

**Giovanni S. Alberti**
MaLGa Center, Department of Mathematics
University of Genoa, Italy
giovanni.alberti@unige.it

**Ernesto De Vito**
MaLGa Center, Department of Mathematics
University of Genoa, Italy
ernesto.devito@unige.it

**Matti Lassas**
Department of Mathematics and Statistics
University of Helsinki, Finland
matti.lassas@helsinki.fi

**Luca Ratti**
MaLGa Center, Department of Mathematics
University of Genoa, Italy
luca.ratti@unige.it

**Matteo Santacesaria**
MaLGa Center, Department of Mathematics
University of Genoa, Italy
matteo.santacesaria@unige.it

## Abstract

In this work, we consider the linear inverse problem $y = Ax + \varepsilon$, where $A\colon X \to Y$ is a known linear operator between the separable Hilbert spaces $X$ and $Y$, $x$ is a random variable in $X$ and $\varepsilon$ is a zero-mean random process in $Y$. This setting covers several inverse problems in imaging including denoising, deblurring and X-ray tomography. Within the classical framework of regularization, we focus on the case where the regularization functional is not given a priori, but learned from data. Our first result is a characterization of the optimal generalized Tikhonov regularizer, with respect to the mean squared error. We find that it is completely independent of the forward operator $A$ and depends only on the mean and covariance of $x$. Then, we consider the problem of learning the regularizer from a finite training set in two different frameworks: one supervised, based on samples of both $x$ and $y$, and one unsupervised, based only on samples of $x$. In both cases we prove generalization bounds, under some weak assumptions on the distribution of $x$ and $\varepsilon$, including the case of sub-Gaussian variables. Our bounds hold in infinite-dimensional spaces, thereby showing that finer and finer discretizations do not make this learning problem harder. The results are validated through numerical simulations.

## 1 Introduction

The aim of an inverse problem is to recover information about a physical quantity from indirect measurements. Virtually all imaging problems and modalities fall within this framework, including denoising, deblurring [14], computed tomography [41] and magnetic resonance imaging [15]. Classical and general approaches to solve inverse problems consist in studying a variational (minimization) problem and can be divided into two classes.

The first is based on the so-called regularization theory [14]. The aim is to recover a single, deterministic, unknown $x^\dagger$ from noisy data $y = F(x^\dagger) + \varepsilon$ by solving a minimization problem

$$\min_x d_Y(F(x), y) + J(x) \tag{1}$$

35th Conference on Neural Information Processing Systems (NeurIPS 2021).

for a fidelity term $d_Y \colon Y \times Y \to \mathbb{R}$ and a regularization functional $J \colon X \to [0, +\infty)$. The latter is chosen in order to mitigate the ill-posedness of the map $F$, and represent some a-priori knowledge on $x$. For instance, in the classical Tikhonov regularization we have $J(x) = \lambda \|x\|_X^2$ for $\lambda > 0$.

The second approach considers the unknown as a random variable and is based on statistical/Bayesian methods [23, 45]. In this case one can recover the unknown using point estimators such as the maximum a posteriori (MAP) estimator, or extract richer information on the probability distribution of the unknown. In practice, the MAP estimator is found by solving a minimization problem of the same form as (1). The main difference is that the fidelity term and the regularizer are tailored to the statistical properties of the unknown and the noise, which are usually assumed to be known.

In recent years, machine learning techniques, and especially deep learning, have shaken the field of inverse problems by providing the basis for data-driven methods that have outperformed the state-of-the-art in most imaging modalities [3, 43]. The most successful methods take inspiration from regularization theory [10, 1, 26, 2, 36, 21, 22, 27, 34, 39, 8, 17, 40]: the physical model given by the forward map $F$ is assumed to be known while the regularizer (or the gradient updates related to it) is learned from a training set. While these approaches have shown impressive results in applications, a solid theory behind their successes is lacking. In view of the many sensitive applications where these methods are already being employed, e.g. in medical imaging, it is of utmost importance to fill this theoretical gap to better understand the strengths and limits of data-driven imaging modalities. Moreover, many inverse problems are naturally formulated in infinite-dimensional spaces [14, 41, 23, 15, 45, 37], and their discretization must be carefully treated due to their ill-posedness [25]. Hence, it is of main interest to provide a theoretical analysis in the infinite-dimensional setting.

In this work, we consider the problem of learning a regularizer for a linear inverse problem in the framework of statistical learning theory, which is the natural setting to derive precise theoretical guarantees. This is part of the growing research area of learning an operator between infinite dimensional spaces [46, 13, 32, 42, 31]. We study the case where the measurements are modeled by a linear, possibly ill-posed, forward map, and the penalty term is a generalized Tikhonov regularizer [47].

More precisely, let $A \colon X \to Y$ be a bounded linear operator between the separable real Hilbert spaces $X$ and $Y$. We consider the inverse problem

$$y = Ax + \varepsilon, \tag{2}$$

which consists of the reconstruction of $x$ from the knowledge of $y$, where $\varepsilon$ represents noise. We assume that $x$ is a random variable on $X$ with mean $\mu$ and covariance $\Sigma_x$, and the noise $\varepsilon$, independent of the variable $x$, is a zero-mean random process on $Y$ with covariance $\Sigma_\varepsilon$, (see Section 2 for more details). The operator $A$ is typically injective but its inverse may be unbounded: typical examples include denoising ($A$ is the identity) and deblurring ($A$ is a convolution operator).

We aim to recover the unknown via generalized Tikhonov regularization. For a quadratic fidelity term $d_Y \colon Y \times Y \to \mathbb{R}$, the minimization problem

$$\min_x d_Y(Ax, y) + \|B^{-1}(x - h)\|_X^2, \tag{3}$$

has a unique solution $R_{h,B}(y)$, called the generalized Tikhonov reconstruction. Here the pair $(h, B)$, where $h \in X$ and $B \colon X \to X$ is a positive bounded operator, is considered as a free parameter that we call *regularization pair*. For example, the operator $B$ can be a smoothing operator in $L^2$, as a negative power of the Laplacian, so that $B^{-1}$ is a differential operator, yielding a classical regularization in Sobolev spaces. We want to characterize and learn the optimal pair $(h, B)$ with respect to the expected, or mean squared, error

$$L(h, B) = \mathbb{E}_{x,y} \|R_{h,B}(y) - x\|_X^2.$$

Our first contribution is a complete characterization of the minimizers of $L$. In particular we find that $(\mu, \Sigma_x^{1/2})$ is a global minimizer (and is unique if $A$ is injective), which shows that the best regularizer is completely independent of the forward operator $A$ and depends only on the mean and covariance of $x$. This is consistent with the known linearized minimum mean squared error estimator in the finite-dimensional case [24], but it is usually not taken into account in the machine learning approaches to inverse problems mentioned above. The extension to the infinite-dimensional case is not straightforward due to the presence of unbounded operators in inverse problems.

Since the computation of the expected error requires the full distribution $\rho$ of $x$ and $y$, we study how this can be approximated from a finite training set. We suppose to have access to a sample of $m$ pairs $\mathbf{z} = \{(x_j, y_j)\}_{j=1}^m$ drawn independently from $\rho$. In view of the results on the expected error, we consider two alternative ways to learn a regularizer pair $(\widehat{h}_{\mathbf{z}}, \widehat{B}_{\mathbf{z}})$ from a training set $\mathbf{z}$: either by minimizing the empirical risk [12] (supervised learning), or by using the empirical mean and covariance of $\{x_j\}$ (unsupervised learning). In both cases, we prove generalization bounds for the sample error $|L(\widehat{h}_{\mathbf{z}}, \widehat{B}_{\mathbf{z}}) - L(\mu, \Sigma_x^{1/2})|$. Under some natural compactness assumptions on the class of regularization pairs, we prove that the sample error has the asymptotic behavior

$$|L(\widehat{h}_{\mathbf{z}}, \widehat{B}_{\mathbf{z}}) - L(\mu, \Sigma_x^{1/2})| \lesssim \frac{1}{\sqrt{m}}, \tag{4}$$

with high probability, in both the supervised and the unsupervised approaches. We stress the point that these bounds hold in the infinite-dimensional setting, or, in other words, they do not depend on the discretization of the signal and of the measurements.

Finally, we complement our theoretical findings with some numerical experiments. For a 1D denoising problem (i.e. $A$ is the identity operator) we replicate the asymptotic bound (4) at different discretization scales. Moreover, we find that the unsupervised approach, despite yielding the same rate (4), clearly outperforms the supervised one.

The paper is organized as follows. In Section 2 we introduce the main notation and technical assumptions that will be used throughout the paper, including several examples. Section 3 presents the main results for the minimization of the expected error, while Section 4 is devoted to the study of the sample error. Numerical experiments are the subject of Section 5. Concluding remarks and discussions are reserved for Section 6.

## 2 Setting the stage

### 2.1 The random objects $x$ and $\varepsilon$

As mentioned in the introduction, we formulate (2) as a statistical inverse problem, where $x$ and $\varepsilon$ are not deterministic but random. Let us start with the description of the prior on $x$.

**Assumption 2.1.** Let $x$ be a random variable on a probability space $(\Omega, \mathbb{P})$ taking values in $X$. More precisely, $x$ is square-integrable, so that its expectation $\mu \in X$ and its covariance $\Sigma_x \colon X \to X$ is a trace-class operator. We assume that $\Sigma_x$ is injective.

Without loss of generality we can always assume that $\Sigma_x$ is injective (i.e., $x$ is non-degenerate), since otherwise it would be enough to consider the inverse problem only in $(\ker \Sigma_x)^\perp \subsetneq X$.

Let us consider some common examples of priors arising in inverse problems.

**Example 2.2** (Gaussian random variables). A general class of priors arises when considering Gaussian random variables. We recall that $x$ is a Gaussian random variable if for all $v \in X$, $\langle x, v \rangle$ is a real Gaussian random variable and, by Fernique's theorem, $x$ is square-integrable [7]. Since $\Sigma_x \colon X \to X$ is self-adjoint, positive and trace-class, we can write its singular value decomposition (SVD) as

$$\Sigma_x v = \sum_k \sigma_k^2 \langle v, e_k \rangle_X e_k, \qquad v \in X,$$

where $\{e_k\}_k$ is an orthonormal basis of $X$, $\sum_k \sigma_k^2 < +\infty$ and $\langle x, e_k \rangle \sim \mathcal{N}(\mu_k, \sigma_k)$, where $\mu = \sum_k \mu_k e_k$ is the mean of $x$. In other words,

$$x = \mu + \sum_k \sigma_k a_k e_k, \tag{5}$$

where $a_k$ are i.i.d. standard Gaussian variables. This shows that, in infinite dimension, since $\sigma_k \to 0$, the variations of $x$ along the direction $e_k$ become smaller and smaller as $k \to +\infty$.

This abstract construction reduces to a smoothness prior by suitably choosing the covariance operator.

**Example 2.3** (Smoothing priors). Let $X = L^2(\mathbb{T}^d)$, where $\mathbb{T}^d = \mathbb{R}^d / \mathbb{Z}^d$ is the $d$-dimensional torus and $d \geq 1$. Let $\Delta$ denote the Laplace-Beltrami operator on $\mathbb{T}^d$, which is simply the classical Laplace

operator on $[0,1]^d$ with periodic boundary conditions. For $s > \frac{d}{2}$, the operator

$$(I - \Delta)^{-s} \colon L^2(\mathbb{T}^d) \to L^2(\mathbb{T}^d)$$

is trace class, and can be used to define the Gaussian distribution $\mathcal{N}(0, (I - \Delta)^{-s})$. In the notation of Example 2.2, the SVD of $(I - \Delta)^{-s}$ is given by $\sigma_k^2 = (1 + 4\pi|k|^2)^{-s}$ and $e_k(t) = e^{2\pi i k \cdot t}$ with $k \in \mathbb{Z}^d$. This enforces a smoothness prior on $x$, depending on the parameter $s$, which controls the decay of the Fourier coefficients of $x$ (see [33, Appendix B] and [7]).

Let us now discuss the model for the noise $\varepsilon$.

**Assumption 2.4.** Let $\varepsilon = (\varepsilon)_{v \in Y}$ be a (linear) random process on $Y$ with zero mean and such that its covariance $\Sigma_\varepsilon \colon Y \to Y$ defined by $\mathbb{E}[\varepsilon_v \varepsilon_w] = \langle \Sigma_\varepsilon v, w \rangle_Y$ is bounded and injective.

Notice that the injectivity of $\Sigma_\varepsilon$ implies that noise is present in all directions of $Y$, whereas the boundedness of $\Sigma_\varepsilon$ allows us to regard the random process $\varepsilon$ as a bounded (linear) operator from $Y$ into $L^2(\Omega, \mathbb{P})$. The reader is referred to [16] and to Appendix A.1 for additional details on random processes in Hilbert spaces. We mention here some basic properties that will be needed for the following discussion.

*Remark* 2.5. Even if $\varepsilon$ may not belong to $Y$ almost surely (see Example 2.7 below), it is always possible to view it as an element of a larger space, as we now discuss. Let $K$ be a separable Hilbert space and $\iota \colon K \to Y$ be an injective linear map such that $\iota(K)$ is dense in $Y$ and

$$\iota^* \circ \Sigma_\varepsilon \circ \iota \colon K \to K^* \text{ is trace-class,}[1] \tag{6}$$

where we identify $Y^* = Y$, but we do not identify $K^*$ with $K$, and we regard $\iota^*$ as the canonical embedding $Y \to K^*$. The restriction of $\varepsilon$ to $K$ is a Hilbert-Schmidt operator from $K$ into $L^2(\Omega, \mathbb{P})$, hence there exists a unique square-integrable random vector $\varepsilon$ taking values in $K^*$ such that $\varepsilon_v = \langle \varepsilon, v \rangle_{K^* \times K}$ for $v \in K$. It is easy to show that the random vector $\varepsilon$ has zero mean and its covariance operator is $\iota^* \circ \Sigma_\varepsilon \circ \iota \colon K \to K^*$, since

$$\mathbb{E}[\langle \varepsilon, v \rangle_{K^* \times K} \langle \varepsilon, w \rangle_{K^* \times K}] = \mathbb{E}[\varepsilon_{\iota(v)} \varepsilon_{\iota(w)}] = \langle \Sigma_\varepsilon \iota v, \iota w \rangle_Y, \qquad v, w \in K. \tag{7}$$

A random variable is always a random process, as we now describe.

**Example 2.6.** A simple example of this abstract construction consists of considering a random variable $\varepsilon$. In this case $\Sigma_\varepsilon$ is trace-class itself, so that we can choose $K = Y$ and $\iota = I$ and $\varepsilon \in Y$ almost surely. As discussed in Example 2.2 for Gaussian variables, this means that, since $\sigma_k \to 0$, the expected amplitude of the noise in the direction $e_k$ goes to 0 as $k \to \infty$. For instance, the choice of $\Sigma_\varepsilon = (I - \Delta)^{-s}$ as in Example 2.3 corresponds to smaller noise levels for higher Fourier modes.

A random process allows for considering noise that is uniformly distributed in all directions.

**Example 2.7** (White noise). The Gaussian white noise $\varepsilon$ is a random process on $Y$ such that for any $v \in Y$ it holds that $\varepsilon_v$ is a standard Gaussian variable (mean 0 and variance 1), so that $\Sigma_\varepsilon = I$. Heuristically, in the notation of Example 2.2, this corresponds to $\sigma_k = 1$ for every $k$, and so by (5)

$$\varepsilon = \sum_k a_k e_k, \qquad a_k \sim \mathcal{N}(0, 1),$$

so that $\varepsilon \notin Y$ with probability 1 whenever $Y$ is infinite dimensional (see, e.g., [16]). In view of Remark 2.5, it is possible to consider a larger space $K^*$ so that $\varepsilon \in K^*$ almost surely. For concreteness of explanation, we focus on the case when $Y = L^2(\mathbb{T}^d)$, a typical framework in imaging. A possible choice for the space $K$ is the Sobolev space $H^s(\mathbb{T}^d)$ with $s > d/2$ (see [25]), so that the canonical embedding $\iota \colon H^s(\mathbb{T}^d) \to L^2(\mathbb{T}^d)$ is a Hilbert-Schmidt operator, hence (6) is satisfied and $\varepsilon$ can naturally be seen as an element of $H^{-s}(\mathbb{T}^d) = H^s(\mathbb{T}^d)^*$.

## 2.2 The new formulation of the inverse problem and of the regularization

As a consequence of Assumptions 2.4, since $\varepsilon$ may not belong to $Y$, the inverse problem (2) must be interpreted from a different perspective, namely considering $y$ as the stochastic process $y_v = \langle Ax, v \rangle_Y + \varepsilon_v$ on $Y$ or by formulating the problem as an equation in $K^*$:

$$y = \iota^* A x + \varepsilon,$$

---

[1] It is worth observing that $K$ and $\iota$ always exist: it is enough to choose them, independently of $\varepsilon$, so that the embedding $\iota \colon K \to Y$ is Hilbert-Schmidt, which implies that $\iota^* \circ \Sigma_\varepsilon \circ \iota$ is trace class, since $\Sigma_\varepsilon$ is bounded.

i.e. $\langle y, v \rangle_{K^* \times K} = \langle Ax, \iota(v) \rangle_Y + \langle \varepsilon, v \rangle_{K^* \times K}$ for $v \in K$, where $\iota^* \colon Y \to K^*$ is the natural embedding. We denote the joint probability distribution of $(x, y)$ on $X \times K^*$ by $\rho$.

We now provide a consistent formulation of the quadratic functional appearing in (3). The goal is to replicate what would be the natural choice in a finite dimensional context, i.e.

$$\min_x \|\Sigma_\varepsilon^{-1/2}(Ax - y)\|_Y^2 + \|B^{-1}(x - h)\|_X^2. \tag{8}$$

Unfortunately, if $Y$ is infinite dimensional, the first factor is in general not well-defined, since for example for Gaussian processes $\varepsilon \in \operatorname{Im} \Sigma_\varepsilon^{1/2}$ with probability 0 [7]. Thus, we need to write this minimization problem in a different formulation. We start by stating the assumptions we make on $B$.

**Assumption 2.8.** Let us assume that $B \colon X \to X$ is a bounded positive operator such that

$$\operatorname{Im}(AB) \subseteq \operatorname{Im}(\Sigma_\varepsilon \iota). \tag{9}$$

It is worth observing that, whenever $Y$ is infinite dimensional, then $\operatorname{Im}(\Sigma_\varepsilon \iota) \subsetneq Y$ since $\Sigma_\varepsilon \iota$ is compact. Furthermore, (9) requires, in some sense, the operator $AB \colon X \to Y$ to be at least as "smoothing" as the operator $\Sigma_\varepsilon \iota \colon K \to Y$. For instance, in the case when $AB$ and $\Sigma_\varepsilon \iota$ have the same left-singular vectors, this condition means that the singular values of $AB$ should go to 0 at least as fast as the singular values of $\Sigma_\varepsilon \iota$.

We are now ready to rewrite the functional in (8). The penalty term involving $B^{-1}(x - h)$ suggests the change of variables $x = h + Bx'$. The corresponding minimization problem for $x'$ reads

$$\min_{x' \in X} \|\Sigma_\varepsilon^{-1/2}(A(h + Bx') - y)\|_Y^2 + \|x'\|_X^2. \tag{10}$$

This expression does not require the injectivity of $B$. By formally expanding the first factor we obtain

$$\|\Sigma_\varepsilon^{-1/2} ABx'\|_Y^2 - 2\langle \Sigma_\varepsilon^{-1/2}(y - Ah), \Sigma_\varepsilon^{-1/2} ABx' \rangle_Y + \|\Sigma_\varepsilon^{-1/2}(y - Ah)\|_Y^2.$$

Let us analyze these three terms separately:

1. Since $\Sigma_\varepsilon$ is self-adjoint we have $\|\Sigma_\varepsilon^{-1/2} ABx'\|_Y^2 = \langle \Sigma_\varepsilon^{-1} ABx', AB'x \rangle_Y$, which is well-defined because $\operatorname{Im}(AB) \subseteq \operatorname{Im}(\Sigma_\varepsilon)$ thanks to (9).

2. The second factor is formally equivalent to $-2\langle y - Ah, \Sigma_\varepsilon^{-1} ABx' \rangle_Y$, which is not well-defined as scalar product in $Y$ since $y$ may not belong to $Y$. However, since $y \in K^*$ and $\Sigma_\varepsilon^{-1} ABx' \in \iota(K)$ by (9), this scalar product can be interpreted as the duality pairing $-2\langle y - \iota^* Ah, \iota^{-1} \Sigma_\varepsilon^{-1} ABx' \rangle_{K^* \times K}$.

3. The third factor $\|\Sigma_\varepsilon^{-1/2}(y - Ah)\|_Y^2$ is independent of $x$, and so it irrelevant for the minimization task: thus, we remove it. This is a key step in infinite dimension, since, as mentioned above, $\|\Sigma_\varepsilon^{-1/2} y\|_Y^2 = +\infty$ almost surely. See [45, Remark 3.8] for additional details on this aspect.

This discussion motivates the introduction of the following functional, formally equivalent to (10). For $y \in K^*$, we define the regularized solution of the inverse problem as $\hat{x} = h + B\hat{x}'$, where

$$\hat{x}' = \arg\min_{x' \in X} \|\Sigma_\varepsilon^{-1/2} ABx'\|_Y^2 - 2\langle y - \iota^* Ah, (\Sigma_\varepsilon \iota)^{-1} ABx' \rangle_{K^* \times K} + \|x'\|_X^2. \tag{11}$$

The minimizer exists and is unique, and gives the following expression for $\hat{x}$:

$$R_{h,B}(y) := h + B\hat{x}' = h + B(BA^* \Sigma_\varepsilon^{-1} AB + I)^{-1}((\Sigma_\varepsilon \iota)^{-1} AB)^*(y - \iota^* Ah), \tag{12}$$

where $R_{h,B} \colon K^* \to X$ is a bounded affine map. See Proposition A.2 for all the details. Note that $((\Sigma_\varepsilon \iota)^{-1} AB)^* \colon K^* \to X$ is well-defined thanks to (9).

## 3 The optimal regularizer

The regularization approach described so far is based on the choice of $h$ and $B$. In classical regularization theory, these are chosen depending on the prior knowledge of the problem under consideration. In the data-driven approach we consider in this work, $h$ and $B$ are learned from training data. In this section, we let learning come into play and consider the problem of determining

the optimal $h$ and $B$, under the assumptions that the distributions of $x$ and $\varepsilon$ are fully known. More precisely, this allows for the explicit computation of the expected error

$$L(h, B) = \mathbb{E}_{(x,y)\sim\rho}\|R_{h,B}(y) - x\|_X^2 = \mathbb{E}_{x,\varepsilon}\|R_{h,B}(\iota^* A x + \varepsilon) - x\|_X^2,$$

which quantifies the mean square error that our regularization functional (11) yields. Optimal choices of $h^\star$ and $B^\star$ are those that minimize $L(h, B)$, and are characterized in the following result.

**Theorem 3.1.** *Let $X$ and $Y$ and be separable real Hilbert spaces, $A\colon X \to Y$ be a bounded linear operator, $x$ and $\varepsilon$ satisfy Assumptions 2.1 and 2.4 and be independent, and $K$ and $\iota$ be as in Remark 2.5. Suppose $B = \Sigma_x^{1/2}$ satisfies Assumption 2.8.*

*Consider the minimization problem*

$$\min_{h,B}\{\mathbb{E}_{(x,y)\sim\rho}\left[\|R_{h,B}(y) - x\|_X^2\right]\}, \tag{13}$$

*where the minimum is taken over all $B$ satisfying Assumption 2.8 and over all $h \in X$. Then $(B^\star, h^\star)$ is a global minimizer of* (13) *if and only if*

$$h^\star = \mu \qquad and \qquad B^2|_{(\ker A)^\perp} = \Sigma_x|_{(\ker A)^\perp}.$$

*In particular, $B^\star = \Sigma_x^{1/2}$ is always a global minimizer, and is unique if $A$ is injective. Furthermore, for every minimizer $(h^\star, B^\star)$, the corresponding reconstruction map is independent of $B^\star$ and, for all $y \in K^*$, is given by*

$$R^\star(y) = \mu + \Sigma_x^{1/2}(\Sigma_x^{1/2}A^*\Sigma_\varepsilon^{-1}A\Sigma_x^{1/2} + I_X)^{-1}((\Sigma_\varepsilon\iota)^{-1}A\Sigma_x^{1/2})^*(y - \iota^*A\mu) \tag{14}$$

$$= \mu + \Sigma_x A^*(\iota^*(A\Sigma_x A^* + \Sigma_\varepsilon))^{-1}(y - \iota^*A\mu). \tag{15}$$

The proof is in Appendix A.3. Some comments on this result are in order.

- By assumption $\iota^*(A\Sigma_x A^* + \Sigma_\varepsilon)$ is an injective compact operator from $Y$ to $K^*$, so that its inverse is not bounded, however it is possible to prove that $\Sigma_x A^*(\iota^*(A\Sigma_x A^* + \Sigma_\varepsilon))^{-1}$ extends to a bounded operator from $K^*$ into $X$. With a slight abuse of notation, we denote this extension in the same way, so that (15) makes sense for all $y \in K^*$.

- To prove this result, we first consider the minimization in (13) over all possibile affine maps, which yields the so-called Linearized Minimum Mean Square Error (LMMSE) estimator of $x$. Then, it is possible to show that such optimal affine functional is of the form $R_{h^\star,B^\star}$, for suitable $B^\star$ and $h^\star$. In a finite-dimensional context, such a result is a direct consequence of the expression of the LMMSE estimator (see, e.g., [24, Theorem 12.1]). Theorem 3.1 generalizes this result to the infinite-dimensional case.

- In the case of Gaussian random variables, the expression of the optimal regularizer $R^\star$ coincides with the maximum a posteriori (MAP) estimator. Nevertheless, our result does not require any assumptions on $x$ and $\varepsilon$ being Gaussian (see the discussion in [19, 20]).

- The minimum expected loss can be computed as

$$L(h^\star, B^\star) = \text{tr}\left(\Sigma_x^{1/2}(\Sigma_x^{1/2}A^*\Sigma_\varepsilon^{-1}A\Sigma_x^{1/2} + I_X)^{-1}\Sigma_x^{1/2}\right), \tag{16}$$

  as it is reported in Appendix A.3.

- It is worth observing that the optimal regularization parameters $B^\star = \Sigma_x^{1/2}$ and $h^\star = \mu$ are independent of $A$ and $\varepsilon$, and depend only on the mean and the covariance of $x$.

## 4 Finding the optimal regularizer: the sample error

The computation of the optimal regularizer proposed in the previous section through the minimization of the expected loss $L$ requires the knowledge of the joint probability distribution $\rho$ of $x$ and $y$. In this section, we suppose that $\rho$ is unknown, [2] but we have access to a sample $\mathbf{z} = \{(x_j, y_j)\}_{j=1}^m$ of $m$ pairs $(x_j, y_j) \in Z = X \times K^*$ drawn independently from the joint probability distribution $\rho$, and we study how to learn an estimator $(\widehat{h}_{\mathbf{z}}, \widehat{B}_{\mathbf{z}})$ of the optimal parameters $(h^\star, B^\star)$. We propose two alternative ways to learn an estimator based on a training sample $\mathbf{z}$. For the ease of notation, from now on we omit the dependence on $\mathbf{z}$.

---

[2]More precisely, we only assume that $\Sigma_\varepsilon$ is known.

1. *Supervised learning*: $(\widehat{h}_S, \widehat{B}_S)$ is determined by minimizing the empirical risk $\widehat{L}$, namely

$$(\widehat{h}_S, \widehat{B}_S) = \underset{(h,B)\in\Theta}{\text{argmin}} \, \widehat{L}(h,B), \qquad \widehat{L}(h,B) = \frac{1}{m}\sum_{j=1}^{m}\|R_{h,B}(y_j) - x_j\|_X^2, \qquad (17)$$

   where $\Theta$ is a suitable subset of $X \times \mathcal{L}(X,X)$.

2. *Unsupervised learning*: since the best parameters are $h^\star = \mu$ and $B^\star = \Sigma_x^{1/2}$, a natural estimator is provided by means of the sample $\{x_j\}$ alone as follows:

$$\widehat{h}_U = \widehat{\mu} = \frac{1}{m}\sum_{j=1}^{m} x_j, \qquad \widehat{B}_U = \widehat{\Sigma}_x^{1/2}, \quad \widehat{\Sigma}_x = \frac{1}{m}\sum_{j=1}^{m}(x_j - \widehat{\mu})\otimes(x_j - \widehat{\mu}). \qquad (18)$$

In both cases, we evaluate the quality of $(\widehat{h}, \widehat{B})$ in terms of its *excess error* $L(\widehat{h}, \widehat{B}) - L(h^\star, B^\star)$.

## 4.1   Supervised learning: empirical risk minimization

There exist several techniques to show the convergence of the empirical risk minimizer to the optimal parameter, involving tools such as the VC dimension and the Rademacher complexity (see, e.g., [44]), which require some compactness assumption on $\Theta$. Here, we fix a Hilbert space $H$ with a compact embedding $j\colon H \to X$ having dense range. For $\varrho_1 > 0$, set

$$\Theta_1 = \{j(\bar{h})\colon \bar{h}\in H, \|\bar{h}\|_H \le \varrho_1\}, \ \Theta_2 = \{j\bar{B}j^*\colon \bar{B}\in \mathrm{HS}(H^*,H), \|\bar{B}\|_{\mathrm{HS}(H^*,H)} \le \varrho_1\}, \quad (19)$$

and define $\Theta$ as the set of pairs $\{(h,B)\in\Theta_1\times\Theta_2 : B \text{ is positive}\}$. Here, $\mathrm{HS}(H^*,H)$ denotes the space of Hilbert-Schmidt operators from $H^*$ to $H$. We further assume that

a) the map $j$ can be decomposed as $j = j_2 \circ j_1$, where $j_1\colon H \to X$ and $j_2\colon X \to X$ are compact and satisfy

$$s_k(j_1) \lesssim k^{-s}, \qquad s > 0, \qquad \text{being } s_k(j_1) \text{ the singular values of } j_1; \qquad (20)$$

   whereas $j_2$ is such that

$$\mathrm{Im}(Aj_2) \subseteq \mathrm{Im}(\Sigma_\varepsilon\iota). \qquad (21)$$

b) The optimal parameter $(h^\star, B^\star) = (\mu, \Sigma_x^{1/2})$ belongs to $\Theta$.

Assumption a), and in particular (20), allows us to explicitly compute the covering numbers, whereas (21) ensures that Assumption 2.8 holds uniformly for each positive operator $B \in \Theta_2$. For example, when $H = H^{\sigma_1}(\mathbb{T}^1)$ and $X = H^{\sigma_2}(\mathbb{T}^1)$ are Sobolev spaces on the one-dimensional torus, assumption a) is satisfied if $s = \sigma_1 - \sigma_2 > 0$. As a consequence, assumption b) can be interpreted as an *a priori* regularity assumption on the problem. Such hypothesis can be relaxed by introducing the approximation error, namely, the rate at which the space $\Theta$ approximates $X \times \mathcal{L}(X,X)$ as the radius $\varrho_1$ grows to $\infty$. Such an analysis, which easily follows from the range-density property of $j$, is not treated here.

Finally, we assume that both the inputs and the outputs are bounded, *i.e.*

$$\mathrm{supp}(\rho) \subset B_Z(\varrho_2), \text{ a ball of } Z = X \times K^* \text{ of radius } \varrho_2. \qquad (22)$$

**Theorem 4.1.** *Under the above conditions, let $(\widehat{h}_S, \widehat{B}_S)$ be defined by* (17) *and take $\tau > 0$. We have*

$$|L(\widehat{h}_S, \widehat{B}_S) - L(h^\star, B^\star)| \le \left(\frac{c_1 + c_2\sqrt{\tau}}{\sqrt{m}}\right)^{1 - \frac{1}{2s'+1}}, \qquad m \ge m_0, \qquad (23)$$

*with probability exceeding $1 - e^{-\tau}$, being $0 < s' < s$, where $c_1, c_2, m_0$ are independent of $m$ and $\tau$.*

The proof of Theorem 4.1 is reported in the Appendices A.4 and A.5. The approach is inspired by [12, Proposition 4] and is suited for a much broader class of learning problems: by adapting Lemma A.6, it is possible to extend the current approach to non-quadratic regularization functionals.

A prominent example of $H$ satisfying (20) comes from Sobolev spaces. Consider, e.g., $X = L^2(\mathbb{T}^d)$, where $\mathbb{T}^d$ is the $d$-dimensional torus, and $H = H^\sigma(\mathbb{T}^d)$. If $\sigma > 0$, the embedding of $H$ in $X$ is compact, and its singular values show a polynomial decay (20) with $s = \sigma/d$.

## 4.2 Unsupervised learning: empirical mean and covariance

As pointed out in (18), it is possible to recover an approximation of the optimal parameter $(h^\star, B^\star)$ only by taking advantage of a sample of the output variable $\{x_j\}_{j=1}^m$. Since this technique does not require matched couples of inputs and outputs, we refer to it as an unsupervised learning approach. In order to prove a bound in probability for the sample error, in this section we assume that $x$ is a $\kappa$-sub-Gaussian random vector, i.e.,

$$\forall v \in X, \langle x, v \rangle_X \text{ is a real sub-Gaussian r.v., i.e. } \|\langle x, v \rangle_X\|_p \leq \kappa\sqrt{p}\|\langle x, v \rangle_X\|_2, \ \forall p > 1, \quad (24)$$

where $\|\langle x, v \rangle\|_p^p = \mathbb{E}[|\langle x, v \rangle|^p]$. It its known [48] that $\mathbb{E}[\|x\|^p]$ is finite for all $p > 0$, so that $x$ has finite mean and its covariance operator $\Sigma_x$ is trace-class. Gaussian random variables are a particular instance of sub-Gaussian random variables by Fernique's theorem [7]. Note that, in infinite-dimensional spaces, bounded random vectors in general are not sub-Gaussian.

We further assume that the injective operator $\Sigma_\varepsilon$ has a bounded inverse, thus $\Sigma_\varepsilon + A\Sigma_x A^*$ is invertible. This is satisfied for example if $\varepsilon$ is the white-noise, since $\Sigma_\varepsilon = I$. We also require that $A^*(\iota^*(\Sigma_\varepsilon + A\Sigma_x A^*))^{-1}$, defined on $\iota^*(Y) \subset K^*$, extends to a bounded operator from $K^*$ into $X$.

**Theorem 4.2.** *Under the above conditions, let* $(\widehat{h}_U, \widehat{B}_U)$ *be defined by* (18) *and take* $\tau > 0$. *Then,*

$$|L(\widehat{h}_U, \widehat{B}_U) - L(h^\star, B^\star)| \leq \frac{c_3 + c_4\sqrt{\tau}}{\sqrt{m}}, \qquad m \geq m_0, \quad (25)$$

*with probability exceeding* $1 - e^{-\tau}$, *where* $m_0$, $c_3$ *and* $c_4$ *depend only on* $\Sigma_x$, $\Sigma_\varepsilon$ *and* $A$.

The proof of Theorem 4.2 is based on several concentration estimates reported in Appendix A.6. The rates we obtain can be meaningfully compared with recent results in supervised learning: see [6, 35].

## 5 Numerical simulations

We report some numerical results obtained from the supervised and unsupervised strategies for a denoising problem, using synthetic data. The goal of these experiments is twofold: on one hand, we want to study the asymptotic properties of the regularizers learned with the techniques proposed in Section 4 as the sample size $m$ grows, verifying Theorems 4.1 and 4.2. On the other hand, we want to assess that those properties, obtained in an infinite-dimensional setting, do not suffer from the curse of dimensionality. We do so by introducing finer and finer discretizations, and showing that the theoretical bounds do not degrade as the dimension of the problem increases.

### 5.1 Problem formulation and discretization

We consider a denoising problem on $X = Y = L^2(\mathbb{T}^1)$, being $\mathbb{T}^1 = \mathbb{R}/\mathbb{Z}$ the one-dimensional torus, which consists in determining a signal $x$ from the noisy measurement $y = x + \varepsilon$ and thus corresponds to the case $A = I$. We define a statistical model both for $\varepsilon$ and for $x$, which we use for the generation of the training data. In the learning process, though, we do not take advantage of the knowledge of the introduced probability distributions, apart from the covariance operator of the noise $\Sigma_\varepsilon$. In accordance with Assumption 2.4, we assume that $\varepsilon$ is a random process on $Y$, and in particular we consider a white noise process, i.e. with zero mean and $\Sigma_\varepsilon = \sigma^2 I$. We consider a noise level of 5%, namely, the standard deviation $\sigma$ is set to the 5% of the peak value of the average signal. In different tests, we employ different white noise processes with different distributions, including the Gaussian (cfr. Example 2.7) and the uniform distributions. Regarding $x$, we assume a Gaussian distribution with fixed mean $\mu$ and covariance $\Sigma_x$, where $\mu = 1 - |2x - 1|$ and $\Sigma_x^{1/2}$ is a convolution operator.

In order to discretize the described problem, we fix $N > 0$ and approximate the space $X$ by means of the $N$−dimensional space generated by a 1D-pixel basis. As a consequence, the functions in $X$ and $Y$ are approximated by vectors in $\mathbb{R}^N$, and the linear operators by matrices in $\mathbb{R}^{N \times N}$. More details on the discretizations and on the random process generation are reported in Appendix A.7.

### 5.2 Implementation and results

We denote $\theta = (h, B)$. The workflow of the numerical experiments is described as follows:

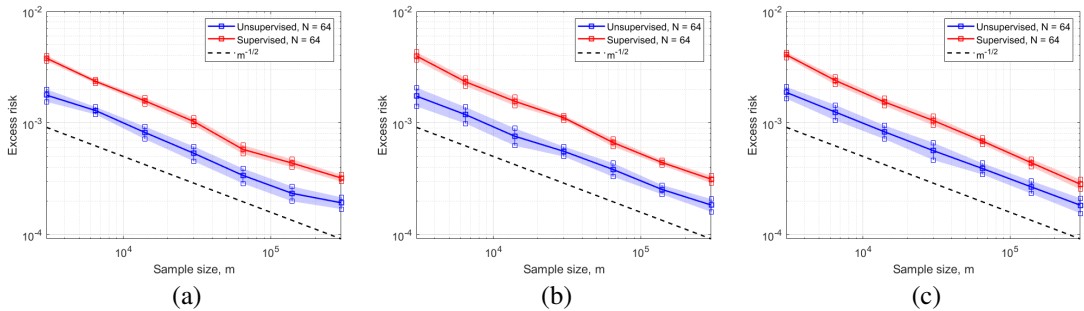

Figure 1: Decay of the excess risks $|L(\widehat{\theta}_S) - L(\theta^\star)|$ and $|L(\widehat{\theta}_U) - L(\theta^\star)|$ in three different cases: Gaussian variable $x$ and (a) Gaussian white noise $\varepsilon$, (b) uniform white noise $\varepsilon$, and (c) white noise $\varepsilon$ uniformly distributed w.r.t. the Haar wavelet transform. We also report standard deviation error bars.

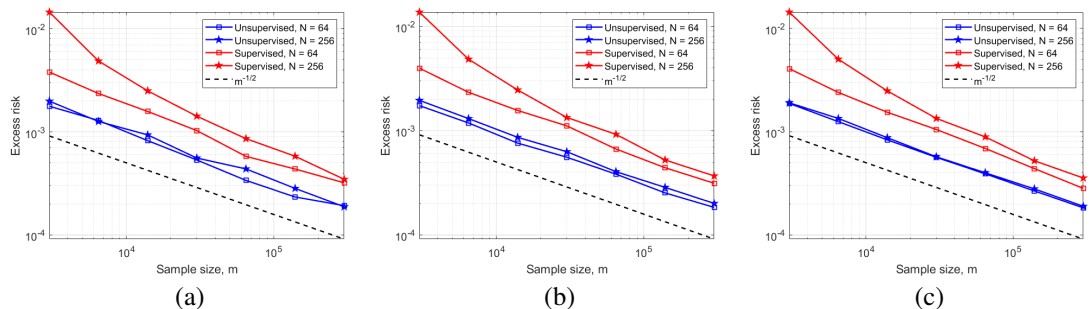

Figure 2: Comparison of decay the excess risks $|L(\widehat{\theta}_S) - L(\theta^\star)|$ and $|L(\widehat{\theta}_U) - L(\theta^\star)|$ with $N = 64$ and $N = 256$, in same three statistical models as in Figure 1.

1. Fix the discretization size $N$, define the optimal regularizer $R_{\theta^\star}$. Compute $L(\theta^\star)$.

2. For each selected value of the sample size $m$,

    - generate the samples $\{x_j\}_{j=1}^m$, $\{\varepsilon_j\}_{j=1}^m$;
    - minimize the empirical risk $\widehat{L}(\theta)$ to find $\widehat{\theta}_S$;
    - compute the empirical mean and covariance $\widehat{\mu}$, $\widehat{\Sigma_x}$ to find $\widehat{\theta}_U$;
    - compute the excess risks $|L(\widehat{\theta}_S) - L(\theta^\star)|$ and $|L(\widehat{\theta}_U) - L(\theta^\star)|$.

3. Show the decay of both computed quantities as $m$ increases.

We compute the mean squared errors $L(\theta^\star)$, $L(\widehat{\theta}_S)$ and $L(\widehat{\theta}_U)$ according to the definition of $L$, thus avoiding the use of a test set. Moreover, we perform the minimization of the empirical risk analytically, thanks to the explicit expression of the regularization functional provided by (12) (see Appendix A.7). As a final remark, the generalization bounds in Theorems 4.1 and 4.2 can be reformulated in expectation. Thus, to verify the expected decay, we repeat the same experiment 30 times, with different training samples for each size $m$ and taking the average in each repetition.[3]

In Figure 1, we present the outcome of the numerical experiments, conducted under different statistical models for $x$ and $\varepsilon$. The sample size ranges between $3 \cdot 10^3$ and $3 \cdot 10^5$. In all the presented scenarios, the decay of the excess risk both in the supervised and in the unsupervised cases agrees with the theoretical estimates, showing a decay of the order $1/\sqrt{m}$. Finally, in Figure 2, we show that the theoretical results are equivalently matched by numerics when the discretization size is increased.

Additional details regarding the results of the numerical experiments are reported in Appendix A.7. Moreover, in Appendix A.8 we replicate the presented numerical study for a different example,

---

[3]All computations were implemented with Matlab R2019a, running on a laptop with 16GB of RAM and 2.2 GHz Intel Core i7 CPU. All the codes are available at `https://github.com/LearnTikhonov/Code`

namely, a deconvolution problem for 1D signals. In this case, $A$ is a convolution operator with respect to a continuous kernel, whose inverse is in general unbounded.

## 6  Conclusions and limitations

We studied the problem of learning a regularization functional for an inverse problem between infinite dimensional spaces. This problem has received huge interest in recent years due to the successes in several imaging modalities. Our work provides theoretical support to machine learning approaches in sensitive applications such as medical imaging.

We have considered the case of a linear inverse problem that is solved via generalized Tikhonov regularization. This involves an unknown operator $B$ and a signal $h$, both to be learned from data. We proposed two learning strategies, one supervised and one unsupervised. Surprisingly, we found that the regularizer learned with the unsupervised strategy has the same (or slightly better) generalization bounds than the supervised one. Furthermore, the unsupervised approach does not need the knowledge of the forward operator $A$ nor that of the distribution of the noise $\varepsilon$. This motivates the development of more advanced unsupervised approaches to the problem, e.g. with deep learning methods (see [26, 27, 34, 36, 39]).

The analysis presented here was possible thanks to the simple form of the regularizer. Our work does not cover, for instance, the case of sparsity promoting regularization functionals [18] or more general convex or non-convex penalty terms arising from deep learning methods. Some results regarding optimal (non-quadratic) regularizers associated with different priors can be found e.g. in [19, 20, 9], which nevertheless deal with a finite-dimensional setting. Extensions to more general regularizers will be the subject of future studies.

## Acknowledgments and Disclosure of Funding

This material is based upon work supported by the Air Force Office of Scientific Research under award number FA8655-20-1-7027. GSA, EDV, LR and MS are members of GNAMPA, INdAM. GSA is supported by a UniGe starting grant "curiosity driven". ML is supported by Academy of Finland, grants 273979 and 284715.

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
