# A Appendix

## A.1 Random processes

The notion of random vector in infinite-dimensional vector spaces is not general enough to describe many models of noise, as for example the white noise described in Example 2.7. To overcome this problem, a possibility is to consider the noise as a *random process* on $Y$ (see the approach in [16]). A random process is a collection $\{\varepsilon_v\}_{v \in Y}$ of real random variables $\varepsilon_v$, each of them defined on the same probability space $(\Omega, \mathcal{F}, \mathbb{P})$ and labelled by vectors $v \in Y$. Here we assume that the random process is linear, with zero mean and bounded. This means that

$$\varepsilon_{\alpha v + \beta w} = \alpha \varepsilon_v + \beta \varepsilon_w, \qquad v, w \in Y,$$

and for each $v \in Y$, $\varepsilon_v$ has zero-mean with finite bounded variance

$$\mathbb{E}[\varepsilon_v^2] \leq C_\varepsilon \|v\|^2, \tag{26}$$

where $C_\varepsilon > 0$ is a suitable constant independent of $v$. The existence of $C_\varepsilon$ is equivalent to assuming that the covariance operator $\Sigma_\varepsilon : Y \to Y$, given by

$$\mathbb{E}[\varepsilon_v \varepsilon_w] = \langle \Sigma_\varepsilon v, w \rangle_Y, \qquad v, w \in Y,$$

is bounded from $Y$ to $Y$. It is easy to show that if $\varepsilon : \Omega \to Y$ is a square-integrable random vector, then the collection of real random variables $\{\varepsilon_v\}_{v \in Y}$

$$\varepsilon_v(\omega) = \langle \varepsilon(\omega), v \rangle_Y \tag{27}$$

is a linear bounded random process. However, the converse is not true as shown by the following example.

**Example A.1.** The Gaussian white noise $\varepsilon$ on $Y$ is a random process such that for any $v \in Y$ it holds that $\varepsilon(v)$ is a zero mean Gaussian variable, and $\Sigma_\varepsilon = I$, i.e., $\mathbb{E}[\varepsilon(v_i)\varepsilon(v_j)] = \langle v_i, v_j \rangle_Y$. Suppose now that $\varepsilon \in Y^*$: then, by Riesz representation theorem there should exist $\hat{\varepsilon} \in Y$ s.t. $\varepsilon(v) = \langle \hat{\varepsilon}, v \rangle_Y$. Nevertheless, this leads to a contradiction, since, letting $\{\phi_i\}_i$ be an orthonormal basis of $Y$,

$$\mathbb{E}[\|\varepsilon\|_{Y^*}^2] = \mathbb{E}[\|\hat{\varepsilon}\|_Y^2] = \sum_{i,j \in \mathbb{N}} \mathbb{E}[\langle \hat{\varepsilon}, \phi_i \rangle_Y \langle \hat{\varepsilon}, \phi_j \rangle_Y] = \sum_{i,j \in \mathbb{N}} \mathbb{E}[\varepsilon(\phi_i)\varepsilon(\phi_j)] = \sum_{i,j \in \mathbb{N}} \langle \phi_i, \phi_j \rangle_Y,$$

which is a divergent sum. It is moreover easy to show that $\mathbb{P}(\|\varepsilon\|_{Y^*} < \infty) = 0$ (see, e.g. [16]).

However, given a random process $\{\varepsilon_v\}_{v \in Y}$, it is always possible to define a Gelfand triple $K \subseteq Y \subseteq K^*$ and random vector $\varepsilon$ taking value in $K^*$ such that (27) holds true for all $v \in K$.

Indeed, let $K$ be a Hilbert space with a continuous embedding $\iota : K \to Y$ such that $\iota(K)$ is dense in $Y$ and the linear map

$$K \ni v \mapsto \varepsilon_{\iota(v)} \in L^2(\Omega, \mathbb{P})$$

is a Hilbert-Schmidt operator. Observe that (26) implies that the linear map

$$Y \ni v \mapsto \varepsilon_v \in L^2(\Omega, \mathbb{P})$$

is always bounded, so that it is enough to assume that $\iota$ is itself a Hilbert-Schmidt operator. To construct the Gelfand triple, we identify $Y^*$ with $Y$, but we do not identify $K^*$ with $K$. Hence, since $\iota(Y)$ is dense in $Y$, then $\iota^* : Y \to K^*$ is injective and $Y$ can be regarded as a (dense) subspace of $K^*$, so that $K \subseteq Y \subseteq K^*$.

For a fixed $\iota$, the canonical identification

$$\mathrm{HS}(K, L^2(\Omega, \mathbb{P})) \simeq L^2(\Omega, \mathbb{P}) \otimes K^* \simeq L^2(\Omega, \mathbb{P}, K^*)$$

implies that there exists $\varepsilon \in L^2(\Omega, \mathbb{P}, K^*)$, *i.e.* a square-integrable random vector $\varepsilon$ in $K^*$, such that

$$\varepsilon_v(\omega) = \langle \varepsilon(\omega), v \rangle_{K^* \times K}$$

almost surely. It is easy to show that the random vector $\varepsilon \in K^*$ has zero mean and its covariance operator $\widetilde{\Sigma}_\varepsilon$, defined as

$$\widetilde{\Sigma}_\varepsilon : K \to K^* \qquad \langle \widetilde{\Sigma}_\varepsilon v, w \rangle_{K^* \times K} = \mathbb{E}[\langle \varepsilon, v \rangle_{K^* \times K} \langle \varepsilon, w \rangle_{K^* \times K}] \qquad v, w \in K,$$

is given by $\widetilde{\Sigma}_\varepsilon = \iota^* \circ \Sigma_\varepsilon \circ \iota$, as shown by (7).

For example, for the white noise on the Hilbert space $Y = L^2(D)$, $D \subset \mathbb{R}^d$, a possible choice is $K = H^s(D)$ with $s > d/2$, so that the embedding is a Hilbert-Schmidt operator (see [25]). Furthermore, it is possible to show that $\varepsilon$ is a Gaussian random vector in $K^*$ and the space $Y \subseteq K^*$ is the corresponding Cameron-Martin space, so that $\mathbb{P}[\varepsilon \in Y] = 0$ (see [7]).

Finally, observe that if the random process is already a square-integrable random vector of $Y$, then $\Sigma_\varepsilon$ is a trace-class operator and we can simply choose $K = Y$. Hence the random process setting extends the usual formalism of random variables. Clearly, if $Y$ is finite dimensional, the two approaches are equivalent.

### A.2 The solution of the regularization problem with fixed $h$ and $B$

Throughout this section and the following ones, we denote the adjoint of an operator between Hilbert spaces $F \colon \mathcal{H}_1 \to \mathcal{H}_2$ as $F^* \colon \mathcal{H}_2^* \to \mathcal{H}_1^*$ such that $\langle F^* u, v \rangle_{\mathcal{H}_1^* \times \mathcal{H}_1} = \langle u, Fv \rangle_{\mathcal{H}_2^* \times \mathcal{H}_2}$ for all $v \in \mathcal{H}_1$, $u \in \mathcal{H}_2^*$. Notice that we identify the spaces $X$ and $Y$ with their dual spaces, so that, e.g., $A^*$ is intended as an operator from $Y$ to $X$. We do not identify $K$ with $K^*$ nor $H$ with $H^*$, so that, e.g., $\iota^* \colon Y \to K^*$.

**Proposition A.2.** *Let*

- *$X$, $Y$ and $K$ be separable real Hilbert spaces;*
- *$A \colon X \to Y$ be a bounded map;*
- *$\Sigma_\varepsilon \colon Y \to Y$ satisfy Assumption 2.4;*
- *$\iota \colon K \to Y$ be an injective linear map satisfying (6);*
- *$B$ satisfy Assumption 2.8 and $h \in X$.*

*For $y \in K^*$, the problem*

$$\hat{x}' = \arg\min_{x' \in X} \|\Sigma_\varepsilon^{-1/2} ABx'\|_Y^2 - 2\langle y - \iota^* Ah, (\Sigma_\varepsilon \iota)^{-1} ABx' \rangle_{K^* \times K} + \|x'\|_X^2 \qquad (28)$$

*admits a unique solution, which is given by the bounded affine function $R'_{h,B} \colon K^* \to X$ defined as*

$$R'_{h,B}(y) = (BA^* \Sigma_\varepsilon^{-1} AB + I)^{-1} ((\Sigma_\varepsilon \iota)^{-1} AB)^* (y - \iota^* Ah). \qquad (29)$$

*Proof.* We have

$$\hat{x}' = \arg\min_{x' \in X} G(x'), \quad G(x') = \|\Sigma_\varepsilon^{-1/2} ABx'\|_Y^2 - 2\langle y - \iota^* Ah, (\Sigma_\varepsilon \iota)^{-1} ABx' \rangle_{K^* \times K} + \|x'\|_X^2.$$

Since the functional $G$ is strictly convex and differentiable, we can find its unique minimum by computing the first-order optimality condition. The Gateaux derivative of $G$ in $x'$ along the direction $w$ reads as

$$\begin{aligned} G'(x')[w] &= 2\langle BA^* \Sigma_\varepsilon^{-1} ABx', w \rangle_X - 2\langle y - \iota^* Ah, (\Sigma_\varepsilon \iota)^{-1} ABw \rangle_{K^* \times K} + 2\langle x', w \rangle_X \\ &= 2\langle BA^* \Sigma_\varepsilon^{-1} ABx', w \rangle_X - 2\langle ((\Sigma_\varepsilon \iota)^{-1} AB)^* (y - \iota^* Ah), w \rangle_X + 2\langle x', w \rangle_X. \end{aligned}$$

Imposing that $G'(\hat{x}')[w] = 0$ for all $w \in X$ therefore implies that

$$(BA^* \Sigma_\varepsilon^{-1} AB + I_X) \hat{x}' = ((\Sigma_\varepsilon \iota)^{-1} AB)^* (y - \iota^* Ah).$$

The operator $BA^* \Sigma_\varepsilon^{-1} AB + I_X$ is invertible since it is a perturbation of the identity by a self-adjoint, non-negative operator, which leads to the expression of the minimizer

$$\hat{x}' = (BA^* \Sigma_\varepsilon^{-1} AB + I_X)^{-1} ((\Sigma_\varepsilon \iota)^{-1} AB)^* (y - \iota^* Ah),$$

as in (29), where we also use that $B$ is self-adjoint since it is positive.

We now show that $R'_{h,B} \colon K^* \to X$ is bounded. We need to show that

$$(BA^* \Sigma_\varepsilon^{-1} AB + I_X)^{-1} (((\Sigma_\varepsilon \iota)^{-1} AB)^* \colon K^* \to X$$

is bounded. As observed above, since $BA^*\Sigma_\varepsilon^{-1}AB$ is self-adjoint and non-negative, we have that $\|(BA^*\Sigma_\varepsilon^{-1}AB + I_X)^{-1}\|_{X \to X} \leq 1$. Thus, it remains to show that $((\Sigma_\varepsilon\iota)^{-1}AB)^* \colon K^* \to X$ is bounded. Recall that this composition is well defined thanks to Assumption 2.8 (see (9)). We prove that

$$(\Sigma_\varepsilon\iota)^{-1}AB \colon X \to K \tag{30}$$

is bounded. By assumption, the map $\Sigma_\varepsilon\iota \colon K \to Y$ is bounded, hence closed. Therefore, $(\Sigma_\varepsilon\iota)^{-1} \colon \iota(K) \subseteq Y \to K$ is closed too. By assumption, $AB \colon X \to Y$ is bounded, hence closed. Thus, the composition (30) is closed, hence bounded thanks to the closed graph theorem. $\qquad\square$

In view of this result, the regularized solution $\hat{x} = h + B\hat{x}'$ to the inverse problem may be written as in (12):

$$\hat{x} = R_{h,B}(y) = W_B y + b_{h,B}, \tag{31}$$

where

$$\begin{aligned} W_B &= B(BA^*\Sigma_\varepsilon^{-1}AB + I_X)^{-1}((\Sigma_\varepsilon\iota)^{-1}AB)^*, \\ b_{h,B} &= h - B(BA^*\Sigma_\varepsilon^{-1}AB + I_X)^{-1}((\Sigma_\varepsilon\iota)^{-1}AB)^*\iota^*Ah, \end{aligned} \tag{32}$$

and $W_B \colon K^* \to X$ is bounded.

We now wish to derive an alternative expression for $W_B$.

**Proposition A.3.** *Assume that the hypotheses of Proposition A.2 hold true. The operator $B^2A^*(\iota^*(AB^2A^* + \Sigma_\varepsilon))^{-1}$ extends to a bounded linear operator from $K^*$ to $X$, which coincides with $W_B$. With an abuse of notation, we have*

$$W_B = B^2A^*(\iota^*(AB^2A^* + \Sigma_\varepsilon))^{-1}. \tag{33}$$

*Proof.* First, observe that

$$((\Sigma_\varepsilon\iota)^{-1}AB)^*\iota^*\Sigma_\varepsilon = ((\Sigma_\varepsilon\iota)^{-1}AB)^*(\Sigma_\varepsilon\iota)^* = ((\Sigma_\varepsilon\iota)(\Sigma_\varepsilon\iota)^{-1}AB)^* = (AB)^* = BA^*,$$

so that

$$((\Sigma_\varepsilon\iota)^{-1}AB)^*\iota^*|_{\mathrm{Im}\,\Sigma_\varepsilon} = BA^*\Sigma_\varepsilon^{-1}. \tag{34}$$

This identity will be used below and in the following.

In order to prove the result, since the operator $\iota^*(AB^2A^* + \Sigma_\varepsilon)$ is injective, it is enough to show that $W_B$ satisfies

$$W_B\iota^*(AB^2A^* + \Sigma_\varepsilon) = B^2A^*.$$

Replacing the expression of $W_B$ given in (32), we obtain

$$B(BA^*\Sigma_\varepsilon^{-1}AB + I_X)^{-1}((\Sigma_\varepsilon\iota)^{-1}AB)^*\iota^*(AB^2A^* + \Sigma_\varepsilon) = B^2A^*.$$

Since $B$ satisfies Assumption 2.8, we have $\mathrm{Im}(AB^2A^*) \subseteq \mathrm{Im}(\Sigma_\varepsilon\iota) \subseteq \mathrm{Im}\,\Sigma_\varepsilon$. Thus, by (34), the above identity is equivalent to

$$B(BA^*\Sigma_\varepsilon^{-1}AB + I_X)^{-1}BA^*\Sigma_\varepsilon^{-1}(AB^2A^* + \Sigma_\varepsilon) = B^2A^*.$$

In order to prove this identity, it is enough to show that

$$(BA^*\Sigma_\varepsilon^{-1}AB + I_X)^{-1}BA^*\Sigma_\varepsilon^{-1}(AB^2A^* + \Sigma_\varepsilon) = BA^*.$$

Since $BA^*\Sigma_\varepsilon^{-1}AB + I_X$ is invertible with bounded inverse, this identity is equivalent to

$$BA^*\Sigma_\varepsilon^{-1}(AB^2A^* + \Sigma_\varepsilon) = (BA^*\Sigma_\varepsilon^{-1}AB + I_X)BA^*.$$

A quick visual inspection shows that this is always true, concluding the proof. $\qquad\square$

## A.3 Proof of Theorem 3.1 and of (16)

The proof of Theorem 3.1 is based on the following observation.

**Lemma A.4.** *Assume that the hypotheses of Theorem 3.1 hold true. Let $B_1, B_2 \colon X \to X$ satisfy Assumption 2.8 and suppose that $B_1$ is injective. Then $W_{B_1} = W_{B_2}$ if and only if*

$$B_1^2 = B_2^2 \quad \text{in } (\ker A)^\perp.$$

*Proof.* Using (32) for $W_{B_1}$ and (33) for $W_{B_2}$, the condition $W_{B_1} = W_{B_2}$ reads

$$B_1(B_1A^*\Sigma_\varepsilon^{-1}AB_1 + I_X)^{-1}((\Sigma_\varepsilon\iota)^{-1}AB_1)^* = B_2^2A^*(\iota^*(AB_2^2A^* + \Sigma_\varepsilon))^{-1}.$$

Since $B_1$ is injective, this is equivalent to

$$((\Sigma_\varepsilon\iota)^{-1}AB_1)^*\iota^*(AB_2^2A^* + \Sigma_\varepsilon) = (B_1A^*\Sigma_\varepsilon^{-1}AB_1 + I_X)B_1^{-1}B_2^2A^*.$$

Since $B_2$ satisfies Assumption 2.8, we have $\mathrm{Im}(AB_2^2A^*) \subseteq \mathrm{Im}(\Sigma_\varepsilon\iota) \subseteq \mathrm{Im}\,\Sigma_\varepsilon$. Thus, by (34),

$$B_1A^*\Sigma_\varepsilon^{-1}(AB_2^2A^* + \Sigma_\varepsilon) = (B_1A^*\Sigma_\varepsilon^{-1}AB_1 + I_X)B_1^{-1}B_2^2A^*.$$

We readily derive

$$B_1A^*\Sigma_\varepsilon^{-1}AB_2^2A^* + B_1A^* = B_1A^*\Sigma_\varepsilon^{-1}AB_2^2A^* + B_1^{-1}B_2^2A^*,$$

yielding $(B_1^2 - B_2^2)A^* = 0$. By continuity of $B_1$ and $B_2$, this is equivalent to having $B_1^2 - B_2^2 = 0$ in $\overline{\mathrm{Im}\,A^*} = (\ker A)^\perp$, as desired. □

*Proof of Theorem 3.1.* Recall that $R_{h,B}$ is given by (31).

*Step 1: arbitrary affine estimators.* We first consider the case of an arbitrary affine estimator $y \mapsto Wy + b$, where $W\colon K^* \to X$ is a bounded linear operator and $b \in X$. Thanks to the independence of $x$ and $\varepsilon$ and the fact that $\mathbb{E}\,x = \mu$ and $\mathbb{E}\,\varepsilon = 0$, the corresponding expected error can be expressed as

$$\begin{aligned}
\mathbb{E}_{x,y}[\|Wy + b - x\|_X^2] &= \mathbb{E}_{x,\varepsilon}[\|(W(\iota^*Ax + \varepsilon) + b - x\|_X^2] \\
&= \mathbb{E}_{x,\varepsilon}[\|(W\iota^*A - I_X)x + W\varepsilon + b\|_X^2] \\
&= \mathbb{E}_x[\|(W\iota^*A - I_X)(x - \mu)\|_X^2] + \|(W\iota^*A - I_X)\mu + b\|_X^2 + \mathbb{E}_\varepsilon[\|W\varepsilon\|_X^2] \\
&= \mathrm{tr}[(W\iota^*A - I_X)\Sigma_x(W\iota^*A - I_X)^*] + \mathrm{tr}[W\iota^*\Sigma_\varepsilon\iota W^*] + \|(W\iota^*A - I_X)\mu + b\|_X^2,
\end{aligned}$$

where the last step is a consequence of the definition of the covariance operators, e.g.

$$\begin{aligned}
\mathbb{E}_\varepsilon[\|W\varepsilon\|_X^2] &= \sum_i \mathbb{E}_\varepsilon[\langle W\varepsilon, \varphi_i\rangle_X^2] \\
&= \sum_i \mathbb{E}_\varepsilon[\langle \varepsilon, W^*\varphi_i\rangle_{K^*\times K}^2] \\
&= \sum_i \langle \Sigma_\varepsilon\iota W^*\varphi_i, \iota W^*\varphi_i\rangle_Y \\
&= \sum_i \langle W\iota^*\Sigma_\varepsilon\iota W^*\varphi_i, \varphi_i\rangle_X \\
&= \mathrm{tr}[W\iota^*\Sigma_\varepsilon\iota W^*],
\end{aligned}$$

where $\{\varphi_i\}$ is an orthonormal basis of $X$ and the third identity follows from (7).

The minimization of the mean square error easily decouples in a minimization in $b$, yielding

$$b = (I_X - W\iota^*A)\mu, \tag{35}$$

and in finding $W$ that minimizes

$$J(W) = \mathrm{tr}\left[(W\iota^*A - I_X)\Sigma_x(W\iota^*A - I_X)^* + W\iota^*\Sigma_\varepsilon\iota W^*\right].$$

It is worth observing that, under the introduced hypotheses, such a functional is well-defined. Indeed, since $\iota^*\Sigma_\varepsilon\iota$ is trace-class (cfr. eq. (6)) and $W$ is a bounded operator, the composition $W\iota^*\Sigma_\varepsilon\iota W^*$ defines a trace-class operator, and analogously with the first term, since $\Sigma_x$ is trace-class.

*Step 2: the optimal B.* Let us consider the minimization of $J$. Note that $J$ is convex and differentiable, hence its minimizer can be found by imposing the following first-order optimality condition

Fix an operator $V : K^* \to X$, by imposing that the Gateaux derivative of $J(W)$ along $V$ is zero, we get

$$\mathrm{tr}\left[V\iota^*(A\Sigma_xA^* + \Sigma_\varepsilon)\iota W^* + W\iota^*(A\Sigma_xA^* + \Sigma_\varepsilon)\iota V^* - V\iota^*A\Sigma_x - \Sigma_xA^*\iota V^*\right] = 0. \tag{36}$$

Choose $V = w \otimes v : K^* \to X$, where $v \in K$ and $w \in X$, then

$$\langle \iota^*(A\Sigma_x A^* + \Sigma_\varepsilon)\iota W^* w, v\rangle_{K^* \times K} + \langle W\iota^*(A\Sigma_x A^* + \Sigma_\varepsilon)\iota v, w\rangle_X = \langle \iota^* A\Sigma_x w, v\rangle_{K^* \times K} + \langle \Sigma_x A^* \iota v, w\rangle_X,$$

so that

$$\langle W\iota^*(A\Sigma_x A^* + \Sigma_\varepsilon)\iota v, w\rangle_X = \langle \Sigma_x A^* \iota v, w\rangle_X.$$

Since $v$ and $w$ are arbitrary, we get

$$W(\iota^* A\Sigma_x A^* \iota + \iota^* \Sigma_\varepsilon \iota) = \Sigma_x A^* \iota. \tag{37}$$

It is easy to show that, if $W$ satisfies (37), equality (36) holds true for all $V$. Since $\iota(K)$ is dense in $Y$ and $W, A, \Sigma_x$ and $\Sigma_\varepsilon$ are bounded, (37) is also equivalent to

$$W\iota^*(A\Sigma_x A^* + \Sigma_\varepsilon) = \Sigma_x A^*. \tag{38}$$

Observe that the operator $A\Sigma_x A^* + \Sigma_\varepsilon \colon Y \to Y$ is positive and injective, hence it has dense range. Further, $\iota$ is injective, and so $\iota^* \colon Y \to K^*$ has dense range. Thus, $\iota^*(A\Sigma_x A^* + \Sigma_\varepsilon)$ has dense range. This shows that there exists at most one bounded operator $W \colon K^* \to X$ satisfying (38). Furthermore, Proposition A.3 gives that $W_{\Sigma_x^{1/2}}$ satisfies (38), so that $W_{\Sigma_x^{1/2}}$ is the unique global minimizer of $J$. Since $\Sigma_x$ is injective, by Lemma A.4 we have that the $B$'s such that $W_B = W_{\Sigma_x^{1/2}}$ are those satisfying $B^2 = \Sigma_x$ in $(\ker A)^\perp$, as desired.

*Step 3: the optimal $h$.* Let us consider (35). It is evident that $b$ is uniquely determined by $W$, and we know that $W = W_{\Sigma_x^{1/2}}$. We now show that, in the case $b = b_{\Sigma_x^{1/2}, h}$ and $W = W_{\Sigma_x^{1/2}}$, equation (35) reduces to $h = \mu$. Indeed, we have

$$h - \Sigma_x^{1/2}(\Sigma_x^{1/2} A^* \Sigma_\varepsilon^{-1} A\Sigma_x^{1/2} + I_X)^{-1}((\Sigma_\varepsilon \iota)^{-1} A\Sigma_x^{1/2})^* \iota^* Ah = (I_X - W_{\Sigma_x^{1/2}} \iota^* A)\mu$$

$$\iff \Sigma_x^{1/2}(\Sigma_x^{1/2} A^* \Sigma_\varepsilon^{-1} A\Sigma_x^{1/2} + I_X)^{-1}((\Sigma_\varepsilon \iota)^{-1} A\Sigma_x^{1/2})^* \iota^* Ah = h - \mu + W_{\Sigma_x^{1/2}} \iota^* A\mu$$

$$\iff ((\Sigma_\varepsilon \iota)^{-1} A\Sigma_x^{1/2})^* \iota^* A(h - \mu) = (\Sigma_x^{1/2} A^* \Sigma_\varepsilon^{-1} A\Sigma_x^{1/2} + I_X)\Sigma_x^{-1/2}(h - \mu)$$

$$\iff \Sigma_x^{1/2} A^* \Sigma_\varepsilon^{-1} A(h - \mu) = \Sigma_x^{1/2} A^* \Sigma_\varepsilon^{-1} A(h - \mu) + \Sigma_x^{-1/2}(h - \mu)$$

$$\iff \Sigma_x^{-1/2}(h - \mu) = 0.$$

Therefore the optimal value is $h^\star = \mu$. $\qquad\qquad\square$

*Proof of* (16). We provide an expression for the minimum value of the expected loss, $L(h^\star, B^\star)$. We have

$$L(h^\star, B^\star) = J(W^\star) = \mathrm{tr}\left[(W^\star \iota^* A - I_X)\Sigma_x(W^\star \iota^* A - I_X)^* + W^\star \iota^* \Sigma_\varepsilon \iota W^{\star *}\right],$$

where the optimal linear functional $W^\star$ satisfies (38), namely, $W^\star \iota^* = \Sigma_x A^* (\Sigma_\varepsilon + A\Sigma_x A^*)^{-1}$.

$$J(W^\star) = \mathrm{tr}\left[(W^\star \iota^*)(A\Sigma_x A^* + \Sigma_\varepsilon)\iota(W^\star)^* - \Sigma_x A^* \iota(W^\star)^* - W^\star \iota^* A\Sigma_x + \Sigma_x\right]$$

$$= \mathrm{tr}\left[\Sigma_x A^* \iota(W^\star)^* - \Sigma_x A^* \iota(W^\star)^* - W^\star \iota^* A\Sigma_x + \Sigma_x\right] =$$

$$= \mathrm{tr}\left[\Sigma_x - \left(\Sigma_x^{1/2}(\Sigma_x^{1/2} A^* \Sigma_\varepsilon^{-1} A\Sigma_x^{1/2} + I_X)^{-1}((\Sigma_\varepsilon \iota)^{-1} A\Sigma_x^{1/2})^* \iota^* A\Sigma_x\right)\right]$$

$$= \mathrm{tr}\left[\Sigma_x^{1/2}\left(I_X - \left((\Sigma_x^{1/2} A^* \Sigma_\varepsilon^{-1} A\Sigma_x^{1/2} + I_X)^{-1}\Sigma_x^{1/2} A^* \Sigma_\varepsilon^{-1} A\Sigma_x^{1/2}\right)\right)\Sigma_x^{1/2}\right]$$

$$= \mathrm{tr}\left(\Sigma_x^{1/2}(\Sigma_x^{1/2} A^* \Sigma_\varepsilon^{-1} A\Sigma_x^{1/2} + I_X)^{-1}\Sigma_x^{1/2}\right),$$

where the second line is a consequence of (38) the third line is due to (32) and the forth line holds true by Assumption 2.8 with $B = \Sigma_x^{1/2}$.

$\qquad\qquad\square$

## A.4   Proof of Theorem 4.1

In order to prove Theorem 4.1, we adapt the classical result on empirical risk minimization to the present discussion, in particular we follow the simple approach in [12]. We postpone to a future work the use of more refined techniques [4, 29]. We consider the parameter space $\Theta \subset X \times \mathcal{L}(X, X)$ as

in (19) and assume in particular that it satisfies (21). We recall that every $B \in \Theta_2$ can be written as $j\bar{B}j^*$, being $\bar{B} \in \mathrm{HS}(H^*, H)$; moreover, since $j = j_2 \circ j_1$, we can also denote it as $B = j_2 \bar{\bar{B}} j_2^*$, where $\bar{\bar{B}}\colon X \to X$, $\bar{\bar{B}} = j_1 \bar{B} j_1^*$. Notice that, using that $j_1$ and $j_2$ are injective and have dense range, given $B$, then $\bar{B}$ and $\bar{\bar{B}}$ are uniquely determined. We can therefore define the following norms on $\Theta$:

$$\|\theta\|_* = \|(h, B)\|_* = \max \left\{ \|h\|_X, \|\bar{\bar{B}}\|_{\mathcal{L}(X,X)} \right\},$$

$$\|\theta\|_{**} = \|(h, B)\|_{**} = \max \left\{ \|\bar{h}\|_H, \|\bar{B}\|_{\mathrm{HS}(H^*,H)} \right\},$$

where $\bar{h} = j^{-1}(h)$. Notice that, according to (19), the set $\Theta$ can be seen as a closed subset of the ball of radius $\varrho_1$ with respect to $\| \cdot \|_{**}$. Nevertheless, the first result we prove does not require that $\Theta$ is chosen as in (19), nor that the functional $R_\theta = R_{h,B}$ is as in (12).

The following result is a restatement of Proposition 4 in [12].

**Lemma A.5.** *Fix a compact subset $\Theta$ of $X \times \{ j_2 \bar{\bar{B}} j_2^* : \bar{\bar{B}}\colon X \to X \text{ bounded} \}$, endowed with the norm $\| \cdot \|_*$, and a family of functions $R_\theta\colon K^* \to X$ labelled by $\theta \in \Theta$ satisfying, for a.e. $(x, y) \in X \times K^*$:*

*a) $\|R_\theta(y) - x\|_X \le M_1$ for every $\theta \in \Theta$;*

*b) $\|R_{\theta_1}(y) - R_{\theta_2}(y)\|_X \le M_2 \|\theta_1 - \theta_2\|_*$, for every $\theta_1, \theta_2 \in \Theta$.*

*Then, with probability $1$ there exist minimizers of $L$ and $\widehat{L}$ over $\Theta$*

$$\theta^\star = \operatorname*{argmin}_{\theta \in \Theta} L(\theta), \qquad \widehat{\theta}_S = \operatorname*{argmin}_{\theta \in \Theta} \widehat{L}(\theta),$$

*and, for all $\eta > 0$,*

$$\mathbb{P}_{\mathbf{z} \sim \rho^m} \left[ |L(\widehat{\theta}_S) - L(\theta^\star)| \le \eta \right] \ge 1 - 2\mathcal{N} \left( \Theta, \frac{\eta}{16 M_1 M_2} \right) e^{-\frac{m\eta^2}{8M_1^4}},$$

*where $\mathcal{N}(\Theta, r)$ denotes the covering number of $\Theta$, i.e., the minimum number of balls of radius $r$ (in norm $\| \cdot \|_*$) whose union contains $\Theta$.*

*Proof.* For $\rho$-almost all $(x, y) \in X \times K^*$

$$\left| \|R_{\theta_1}(y) - x\|_X^2 - \|R_{\theta_2}(y) - x\|_X^2 \right| = |\langle R_{\theta_1}(y) - R_{\theta_2}(y), R_{\theta_1}(y) - x + R_{\theta_2}(y) - x \rangle|$$
$$\le 2M_1 M_2 \|\theta_1 - \theta_2\|_*.$$

By integrating with respect to the probability distribution $\rho$ or the empirical measure $\widehat{\rho}$, the above bound holds for $L, \widehat{L}$. Indeed,

$$|L(\theta_1) - L(\theta_2)| = \left| \mathbb{E}[\|R_{\theta_1}(y) - x\|_X^2] - \mathbb{E}[\|R_{\theta_2}(y) - x\|_X^2] \right|$$
$$\le \mathbb{E}\left[ \left| \|R_{\theta_1}(y) - x\|_X^2 - \|R_{\theta_2}(y) - x\|_X^2 \right| \right] \le 2M_1 M_2 \|\theta_1 - \theta_2\|_*, \tag{39}$$

and, with probability $1$,

$$|\widehat{L}(\theta_1) - \widehat{L}(\theta_2)| = \left| \frac{1}{m} \sum_{j=1}^m \|R_{\theta_1}(y_j) - x_j\|_X^2 - \frac{1}{m} \sum_{j=1}^m \|R_{\theta_2}(y_j) - x_j\|_X^2 \right|$$
$$\le \frac{1}{m} \sum_{j=1}^m \left| \|R_{\theta_1}(y_j) - x_j\|_X^2 - \|R_{\theta_2}(y_j) - x_j\|_X^2 \right| \tag{40}$$
$$\le 2M_1 M_2 \|\theta_1 - \theta_2\|_*.$$

Since both $L$ and $\widehat{L}$ are Lipschitz continuous and $\Theta$ is compact, the corresponding minimizers $\theta^\star$ and $\widehat{\theta}_S$ exist almost surely.

Next, we notice that the event $\{|L(\widehat{\theta}_S) - L(\theta^\star)| \le \eta\}$ is a superset of the event $\left\{ \sup_{\theta \in \Theta} |\widehat{L}(\theta) - L(\theta)| \le \eta/2 \right\}$. Indeed,

$$\sup_{\theta \in \Theta} |\widehat{L}(\theta) - L(\theta)| \le \frac{\eta}{2} \quad \Rightarrow \quad L(\widehat{\theta}_S) - \widehat{L}(\widehat{\theta}_S) \le \frac{\eta}{2} \quad \text{and} \quad \widehat{L}(\theta^\star) - L(\theta^\star) \le \frac{\eta}{2},$$

and ultimately it also holds that

$$0 \leq L(\widehat{\theta}_S) - L(\theta^\star) = \left(L(\widehat{\theta}_S) - \widehat{L}(\widehat{\theta}_S)\right) + \left(\widehat{L}(\widehat{\theta}_S) - \widehat{L}(\theta^\star)\right) + \left(\widehat{L}(\theta^\star) - L(\theta^\star)\right) \leq \eta,$$

where we also used the fact that the central difference is negative by definition of $\widehat{\theta}_S$. Thus,

$$\mathbb{P}_{\mathbf{z}\sim\rho^m}\left[|L(\widehat{\theta}_S) - L(\theta^\star)| \leq \eta\right] \geq \mathbb{P}_{\mathbf{z}\sim\rho^m}\left[\sup_{\theta\in\Theta}|\widehat{L}(\theta) - L(\theta)| \leq \frac{\eta}{2}\right].$$

We now provide a lower bound for the latter term. In view of (39) and (40), by using the reverse triangle inequality, for every $\theta_1, \theta_2 \in \Theta$,

$$\left||\widehat{L}(\theta_1) - L(\theta_1)| - |\widehat{L}(\theta_2) - L(\theta_2)|\right| \leq 4M_1 M_2 \|\theta_1 - \theta_2\|_*.$$

Let now $N = \mathcal{N}\left(\Theta, \frac{\eta}{8M_1 M_2}\right)$ and consider a discrete set $\theta_1, \ldots, \theta_N$ such that the balls $B_k$ centered at $\theta_k$ with radius $\frac{\eta}{8M_1 M_2}$ cover the entire $\Theta$. In each ball $B_k$, for every $\theta \in B_k$ it holds

$$\left||\widehat{L}(\theta) - L(\theta)| - |\widehat{L}(\theta_k) - L(\theta_k)|\right| \leq 4M_1 M_2 \|\theta - \theta_k\|_* \leq \frac{\eta}{2}.$$

Therefore, the event $|\widehat{L}(\theta) - L(\theta)| > \eta$ is a subset of $|\widehat{L}(\theta_k) - L(\theta_k)| > \frac{\eta}{2}$, and a bound (in probability) of this term can be provided by standard concentration results. Indeed, $\widehat{L}(\theta_k)$ is the sample average of $m$ realization of the random variable $\|R_{\theta_k}(y) - x\|_X^2$, whose expectation is $L(\theta_k)$. Moreover, such random variable is bounded by $M_1^2$ by assumption, and therefore via Hoeffding's inequality

$$\mathbb{P}_{\mathbf{z}\sim\rho^m}\left[\sup_{\theta\in B_k}|\widehat{L}(\theta) - L(\theta)| > \eta\right] \leq \mathbb{P}_{\mathbf{z}\sim\rho^m}\left[|\widehat{L}(\theta_k) - L(\theta_k)| > \frac{\eta}{2}\right] \leq 2e^{-\frac{m\eta^2}{2M_1^4}}.$$

Notice that this inequality holds uniformly in $k$. Finally, since $\Theta$ is covered by the union of the balls $B_1, \ldots, B_N$, with $N = \mathcal{N}\left(\Theta, \frac{\eta}{8M_1 M_2}\right)$, we finally obtain

$$\begin{aligned}
\mathbb{P}_{\mathbf{z}\sim\rho^m}\left[\sup_{\theta\in\Theta}|\widehat{L}(\theta) - L(\theta)| \leq \eta\right] &= 1 - \mathbb{P}_{\mathbf{z}\sim\rho^m}\left[\sup_{\theta\in\Theta}|\widehat{L}(\theta) - L(\theta)| > \eta\right] \\
&\geq 1 - \sum_{k=1}^{N} \mathbb{P}_{\mathbf{z}\sim\rho^m}\left[\sup_{\theta\in B_k}|\widehat{L}(\theta) - L(\theta)| > \eta\right] \\
&\geq 1 - 2Ne^{-\frac{m\eta^2}{2M_1^4}}. \qquad \square
\end{aligned}$$

Lemma A.5 provides a very general result: in order to apply it to our current framework, we have to first show that the functional $R_\theta$ defined as in (12) satisfies the assumptions $a)$ and $b)$ in the statement. This is the subject of the following result.

**Lemma A.6.** *Under the assumptions of Section 4.1, let $\Theta$ be as in* (19)*. then the family of functions $R_\theta \colon K^* \to X$, defined by* (12)*, satisfies the assumptions of Lemma A.5.*

*Proof.* Without loss of generality we assume that $\|j_1\|_{\mathcal{L}(H,X)} \leq 1$ and $\|j_2\|_{\mathcal{L}(X,X)} \leq 1$. We first notice that, thanks to (21), for any $B_1, B_2 \in \Theta$,

$$\begin{aligned}
\|((\Sigma_\varepsilon\iota)^{-1}AB_1)^* - ((\Sigma_\varepsilon\iota)^{-1}AB_2)^*\|_{\mathcal{L}(K^*,X)} &\leq \|(\Sigma_\varepsilon\iota)^{-1}A(B_1 - B_2)\|_{\mathcal{L}(X,K)} \\
&= \|(\Sigma_\varepsilon\iota)^{-1}Aj_2(\bar{B}_1 - \bar{B}_2)j_2^*\|_{\mathcal{L}(X,K)} \\
&\leq \|(\Sigma_\varepsilon\iota)^{-1}Aj_2\|_{\mathcal{L}(X,K)}\|\bar{\bar{B}}_1 - \bar{\bar{B}}_2\|_{\mathcal{L}(X,X)}.
\end{aligned}$$

Denote by $\varrho_3 = \|(\Sigma_\varepsilon\iota)^{-1}Aj_2\|_{\mathcal{L}(X,K)}$. Note that, thanks to (21), arguing as in the proof of Proposition A.2 we have that $(\Sigma_\varepsilon\iota)^{-1}Aj_2 \colon X \to K$ is bounded. Then, by (19), for every $B_1, B_2 \in \Theta_2$,

$$\begin{aligned}
\|((\Sigma_\varepsilon\iota)^{-1}AB_1)^* - ((\Sigma_\varepsilon\iota)^{-1}AB_2)^*\|_{\mathcal{L}(X,K)} &\leq \varrho_3\|\bar{\bar{B}}_1 - \bar{\bar{B}}_2\|_{\mathcal{L}(X,X)}, \\
\|((\Sigma_\varepsilon\iota)^{-1}AB_1)^*\|_{\mathcal{L}(X,K)} &\leq \varrho_1\varrho_3.
\end{aligned} \tag{41}$$

Assumption $a)$ in Lemma A.5 requires that $\|R_\theta(y) - x\|_X \le M_1$ for a.e. $(x, y) \in X \times K^*$ and for all $\theta$. Notice that, by the expression of $R_\theta = R_{h,B}$ in (12),

$$\|R_\theta(y) - x\|_X \le \|h\|_X + \|B\|_{\mathcal{L}(X,X)}\|(BA^*\Sigma_\varepsilon^{-1}AB + I_X)^{-1}\|_{\mathcal{L}(X,X)}\|((\Sigma_\varepsilon\iota)^{-1}AB)^*\|_{\mathcal{L}(K^*,X)}$$
$$\cdot(\|y\|_{K^*} + \|\iota^*A\|_{\mathcal{L}(X,K^*)}\|h\|_X) + \|x\|_X$$
$$\le \varrho_1 + \varrho_1^2\varrho_3(\varrho_2 + \|\iota^*A\|_{\mathcal{L}(X,K^*)}\varrho_1) + \rho_2 =: M_1,$$

where we have also used (19), (22), (41) and the fact that the norm of $(BA^*\Sigma_\varepsilon^{-1}AB + I_X)^{-1}$ is less than or equal to 1.

Assumption $b)$ requires instead that $\|R_{\theta_1}(y) - R_{\theta_2}(y)\|_X \le M_2\|\theta_1 - \theta_2\|_*$. According to the definition of $\|\cdot\|_*$, we can decouple the perturbation of $\theta$ and study separately the perturbation of $h$ and of $B$. We observe that

$$R_{h_1,B}(y) - R_{h_2,B}(y) = (h_1 - h_2) - B(BA^*\Sigma_\varepsilon^{-1}AB + I_X)^{-1}((\Sigma_\varepsilon\iota)^{-1}AB)^*\iota^*A(h_1 - h_2),$$

hence again by (41), (22) and (19) we get

$$\|R_{h_1,B}(y) - R_{h_2,B}(y)\| \le (1 + \varrho_1^2\varrho_3\|\iota^*A\|_{\mathcal{L}(X,K^*)})\|h_1 - h_2\|_X.$$

The treatment of the perturbations of $B$ is slightly more delicate. Let $C_i = (B_iA^*\Sigma_\varepsilon^{-1}AB_i + I_X)^{-1}$. Then we have

$$R_{h,B_1}(y) - R_{h,B_2}(y)$$
$$= (B_1C_1((\Sigma_\varepsilon\iota)^{-1}AB_1)^* - B_2C_2((\Sigma_\varepsilon\iota)^{-1}AB_2)^*)(y - \iota^*Ah)$$
$$= (B_1 - B_2)C_1((\Sigma_\varepsilon\iota)^{-1}AB_1)^*(y - \iota^*Ah) + B_2(C_1 - C_2)((\Sigma_\varepsilon\iota)^{-1}AB_1)^*(y - \iota^*Ah)$$
$$+ B_2C_2(((\Sigma_\varepsilon\iota)^{-1}AB_1)^* - ((\Sigma_\varepsilon\iota)^{-1}AB_2)^*)(y - \iota^*Ah)$$

In the latter summation, by means of (41), (22) and (19) we easily get that the first and the third terms are both bounded by $\varrho_1\varrho_3(\varrho_2 + \|\iota^*A\|_{\mathcal{L}(X,K^*)}\varrho_1)\|\bar{\bar{B}}_1 - \bar{\bar{B}}_2\|_{\mathcal{L}(X,X)}$. The second term can be reformulated taking into account that

$$C_1 - C_2 = (I_X + B_1A^*\Sigma_\varepsilon^{-1}AB_1)^{-1} - (I_X + B_2A^*\Sigma_\varepsilon^{-1}AB_2)^{-1}$$
$$= (I_X + B_1A^*\Sigma_\varepsilon^{-1}AB_1)^{-1}(B_2A^*\Sigma_\varepsilon^{-1}AB_2 - B_1A^*\Sigma_\varepsilon^{-1}AB_1)(I_X + B_2A^*\Sigma_\varepsilon^{-1}AB_2)^{-1},$$

and its norm can be bounded by $2\varrho_1^3\varrho_3^2\|\iota^*A\|_{\mathcal{L}(X,K^*)}(\varrho_2 + \|\iota^*A\|_{\mathcal{L}(X,K^*)}\varrho_1)\|\bar{\bar{B}}_1 - \bar{\bar{B}}_2\|_{\mathcal{L}(X,X)}$ using similar arguments. $\square$

Now that the assumptions $a)$ and $b)$ of Lemma A.5 are guaranteed, we have to show the compactness of the parameter class $\Theta$. The following lemma only assumes that $\Theta$ is defined as in (19), by means of a Hilbert space $H$ and a compact, dense-range operator $j\colon H \to X$.

**Lemma A.7.** *The set $\Theta$ defined as in* (19) *is a compact subset of $X \times \{j_2\bar{\bar{B}}j_2^* : \bar{\bar{B}}\colon X \to X \text{ bounded}\}$ with respect to the topology induced by the norm $\|\cdot\|_*$.*

*Proof.* We first show that $\Theta_1 \times \Theta_2$ is compact. Set

$$\bar{\bar{\Theta}}_2 = \{j_1\bar{B}j_1^* : \bar{B} \in \mathrm{HS}(H^*, H), \|\bar{B}\|_{\mathrm{HS}(H^*,H)} \le \varrho_1\} \tag{42}$$

so that $\Theta_2 = \{j_2\bar{\bar{B}}j_2^* : \bar{\bar{B}} \in \bar{\bar{\Theta}}_2\}$. The definition of the norm $\|\cdot\|_*$ implies that $\Theta_1 \times \Theta_2$ is compact with respect to the topology induced by the norm $\|\cdot\|_*$ if and only if $\Theta_1 \times \bar{\bar{\Theta}}_2$ is compact as subset of $X \times \mathrm{HS}(X, X)$ endowed with the product topology. Hence, it is enough to show that $\Theta_1$ and $\bar{\bar{\Theta}}_2$ are compact in $X$ and $\mathrm{HS}(X, X)$, respectively. By definition, since $j$ is compact and $\Theta_1$ is the image of the closed ball of radius $\rho_1$ in $H$, then $\Theta_1$ is compact.

In order to prove that $\bar{\bar{\Theta}}_2$ is compact, we identify $\mathrm{HS}(H^*, H)$ and $\mathrm{HS}(X, X)$ with $H \otimes H$ and $X \otimes X$, respectively, so that for all $v, w \in H$, $v \otimes w\colon H^* \to H$ is the rank one operator

$$(v \otimes w)(z) = \langle z, w\rangle_{H^*,H}\, v, \qquad z \in H^*.$$

With this identification, since

$$j_1(v \otimes w)j_1^* = (j_1v) \otimes (j_1w),$$

the map $\bar{B} \mapsto j_1 \bar{B} j_1^*$ is given by

$$j_1 \otimes j_1 : H \otimes H \to X \otimes X,$$

which is compact, since $j_1$ is so. As above, $\bar{\bar{\Theta}}_2$ is the image of the closed ball of radius $\rho_1$ in $H \otimes H$, so that it is a compact subset of $\mathrm{HS}(X, X)$.

The compactness of $\Theta$ follows from the fact that the subset of positive operators $\bar{\bar{B}} : X \to X$ is closed in $\mathrm{HS}(X, X)$ and $B = j_2 \bar{\bar{B}} j_2^*$ is positive if and only if $\bar{\bar{B}}$ is positive.

$$\square$$

To conclude the proof of Theorem 4.1, we need to provide an explicit expression for the covering numbers of the set $\Theta$ in the $\| \cdot \|_*$ norm. This is possible, e.g. by assuming the polynomial decay of the singular values of $j_1$ as in (20), by means of some tools that are presented in the next section.

## A.5   Entropy numbers, singular values and covering numbers

Let $\mathcal{H}$ and $\mathcal{X}$ be real Hilbert spaces and let $\mathcal{B}$ denote the unit closed ball in $\mathcal{H}$. We use instead the notation $B(v, \varepsilon)$ to denote the closed ball in $\mathcal{X}$ with center $v$ and radius $\varepsilon$. For any compact operator $T : \mathcal{H} \to \mathcal{X}$ we can define the following quantities.

1. **Entropy numbers:** for each $k \in \mathbb{N}$, $k \geq 1$,

$$\varepsilon_k(T) = \inf\{\varepsilon > 0 \mid \exists v_1, \ldots, v_k \in \mathcal{X} \text{ such that } \cup_{i=1}^k B(v_i, \varepsilon) \supseteq T(\mathcal{B})\};$$

2. **Singular values:** $s_k(T) = \lambda_k(|T|)$, where $\lambda_k(|T|)$ is the $k$-th non-zero eigenvalue of $|T|$, which are counted with their multiplicity and ordered in a non-increasing way. If $|T|$ has less than $K$ non-zero eigenvalues, then $s_k(T) = 0$ for $k \geq K$.

3. **Covering numbers** of $T$: the covering numbers of the set $T(\mathcal{B})$; namely, for $r > 0$,

$$\mathcal{N}_r(T) = \mathcal{N}(T(\mathcal{B}), r) = \inf\{k \in \mathbb{N}_* \mid \exists v_1, \ldots, v_k \in \mathcal{X} \text{ such that } \cup_{i=1}^k B(v_i, r) \supseteq T(\mathcal{B})\}.$$

Properties of covering and entropy numbers have recently been used in the study of instability in inverse problems [28]. We have the following results (see [11]):

$$\varepsilon_k(T) \leq r \qquad \Longleftrightarrow \qquad \mathcal{N}_r(T) \leq k; \tag{43}$$

and for all $k \in \mathbb{N}$, $k \geq 1$

$$\sup_{1 \leq \ell < \infty} \left( k^{-1/\ell} \left( \Pi_{i=1}^\ell s_i(T) \right)^{1/\ell} \right) \leq \varepsilon_k(T) \leq 14 \sup_{1 \leq \ell < \infty} \left( k^{-1/\ell} \left( \Pi_{i=1}^\ell s_i(T) \right)^{1/\ell} \right). \tag{44}$$

We now use these properties to quantify the covering numbers $\mathcal{N}(\Theta, r)$ appearing in Lemma A.5. We assume, for simplicity, that $\varrho_1 = 1$. For $\varrho_1 \neq 1$, we can rescale the covering numbers by the formula $\mathcal{N}(\varrho B, r) = \mathcal{N}(B, r/\varrho)$, where $\varrho B = \{\varrho b : b \in B\}$. By the definition of $\Theta$, it is evident that $\mathcal{N}(\Theta, r) \leq \mathcal{N}(\Theta_1, r)\mathcal{N}(\Theta_2, r)$. The following two lemmas take care of estimating the covering numbers of $\Theta_1$ and $\Theta_2$, respectively.

**Lemma A.8.** *Under Assumption* (20) *we have*

$$\ln(\mathcal{N}(\Theta_1, r)) \leq C r^{-\frac{1}{s}}, \qquad r > 0, \tag{45}$$

*where $C > 0$ is independent of $r$.*

*Proof.* Observe that $\mathcal{N}(\Theta_1, r) = \mathcal{N}_r(j) \leq \mathcal{N}_{\frac{r}{\|j_2\|_{\mathcal{L}(X,X)}}}(j_1)$. Condition (20) yields

$$\Pi_{i=1}^\ell s_i(j_1) \lesssim (\ell!)^{-s} \lesssim \ell^{-s\ell} e^{s\ell},$$

where the last bound is a consequence of the fact that $(\ell)! \geq e \ell^\ell e^{-\ell}$. Estimate (44) implies that

$$\varepsilon_k(j_1) \lesssim \sup_{1 \leq \ell < \infty} \left( k^{-1/\ell} \left( \ell^{-s\ell} e^{s\ell} \right)^{1/\ell} \right) = \sup_{1 \leq \ell < \infty} \left( k^{-1/\ell} \ell^{-s} e^s \right).$$

Let $t = 1/\ell$, since the function $e^{-t \ln k} t^s$ takes its maximum at $t = s/\ln k$, then

$$\varepsilon_k(j_1) \lesssim (\ln k)^{-s},$$

where the constant in $\lesssim$ depends on $s$. Eq. (43) yields $\mathcal{N}_{(\ln k)^{-s}}(j_1) \lesssim k$ and ultimately the thesis.   $\square$

**Lemma A.9.** *Under assumption* (20), *for every* $s' \in (0, s)$ *we have*

$$\ln(\mathcal{N}(\Theta_2, r)) \leq Cr^{-\frac{1}{s'}}, \qquad r > 0, \tag{46}$$

*where* $C > 0$ *is independent of* $r$.

*Proof.* Observe that $\mathcal{N}(\Theta_2, r) = \mathcal{N}(\bar{\bar{\Theta}}_2, r) = \mathcal{N}_r(j_1 \otimes j_1)$, since $j_1 \otimes j_1$ represents the (compact) embedding of $\mathrm{HS}(H^*, H)$ into $\mathrm{HS}(X, X)$, see the proof of Lemma A.7 and (42).

We bound the singular values of $j_1 \otimes j_1$. Let $f \colon (0, +\infty) \to \mathbb{N}$ be defined by

$$f(t) = \#\{(k_1, k_2) \in \mathbb{N}_* \times \mathbb{N}_* : s_{k_1, k_2}(j_1 \otimes j_1) = s_{k_1}(j_1) s_{k_2}(j_1) \geq t\}.$$

Then, for all $k \in \mathbb{N}_*$ we have

$$s_k(j_1 \otimes j_1) = \sup\{t \in (0, +\infty) : f(t) \geq k\}.$$

By the polynomial decay of the singular values (20)

$$f(t) \leq \#\{(k_1, k_2) \in \mathbb{N}_* \times \mathbb{N}_* : (k_1 k_2)^{-s} \geq Ct\} = g(t),$$

where $C$ is a suitable constant. Then,

$$s_k(j_1 \otimes j_1) \leq \sup\{t \in (0, +\infty) : g(t) \geq k\}.$$

We now estimate $g(t)$. Let $\tau = (Ct)^{-1/s}$ and $N = [\tau]$ be the integer part of $\tau$. Fix $0 < \varepsilon < 1$

$$g(t) = \#\{(k_1, k_2) \in \mathbb{N}_* \times \mathbb{N}_* : k_1 k_2 \leq (Ct)^{-1/s} = \tau\}$$

$$= \sum_{k_1=1}^{N} \left[\frac{\tau}{k_1}\right] \leq \sum_{k_1=1}^{N} \frac{\tau}{k_1} \leq \tau + \int_1^N \frac{\tau}{x} dx$$

$$\leq \tau(1 + \ln(\tau)) \leq \frac{1}{\varepsilon} \tau^{1+\varepsilon} \lesssim \frac{1}{\varepsilon} t^{-(1+\varepsilon)/s}$$

where $(1 + \ln x) \leq \varepsilon^{-1} x^\varepsilon$ provided that $0 < \varepsilon < 1$. Set $s/2 \leq s' = s/(1 + \varepsilon) < s$, then

$$g(t) \lesssim \frac{s}{s - s'} t^{-\frac{1}{s'}}$$

so that

$$s_k(j_1 \otimes j_1) \lesssim k^{-s'}. \tag{47}$$

Clearly, the above bound holds true also when $0 < s' < s/2$. The proof follows by repeating the argument of the proof of Lemma A.8. $\qquad\square$

We are now able to prove the main result of section 4.1.

*Proof of Theorem 4.1.* By Lemma A.8 and A.9, we conclude that

$$\ln(\mathcal{N}(\Theta, r)) \leq \ln(\mathcal{N}(\Theta_1, r)) + \ln(\mathcal{N}(\Theta_2, r)) \lesssim r^{-\frac{1}{s}} + r^{-\frac{1}{s'}} \lesssim r^{-\frac{1}{s'}}.$$

Substituting this result in Lemma A.5 allows to conclude that

$$\mathbb{P}_{\mathbf{z} \sim \rho^m}\left[|L(\widehat{\theta}_{\mathbf{z}}) - L(\theta^*)| \leq \eta\right] \geq 1 - e^{\tilde{c}_1 \eta^{-1/s'} - \tilde{c}_2 m \eta^2} = 1 - e^{-\tau}.$$

We can express $\eta$ as a function of $m$ and $\tau$ when $\eta < 1$ by the following estimate:

$$\tau \geq \tilde{c}_2 m \eta^2 - \tilde{c}_1 \eta^{-1/s'} \quad \Rightarrow \quad \tilde{c}_2 m \eta^{2 + \frac{1}{s'}} \leq \tilde{c}_1 + \tau \eta^{1/s'} \leq \tilde{c}_1 + \tau,$$

and therefore (with constants $c_1, c_2$ independent of $m, \tau, \eta$)

$$|L(\widehat{\theta}_{\mathbf{z}}) - L(\theta^*)| \leq \eta \leq \left(\frac{\tilde{c}_1 + \tau}{\tilde{c}_2 m}\right)^{\frac{1}{2+1/s'}} \leq \left(\frac{c_1 + c_2 \sqrt{\tau}}{\sqrt{m}}\right)^{1 - \frac{1}{2s'+1}},$$

with probability larger than or equal to $1 - e^{-\tau}$. $\qquad\square$

## A.6 Proof of Theorem 4.2

In the unsupervised setting, the regularizer $R_{\widehat{h}_U, \widehat{B}_U}$ is given by

$$R_{\widehat{h}_U, \widehat{B}_U}(y) = \widehat{h}_U + \widehat{B}_U^2 A^* (\iota^* (A\widehat{B}_U^2 A^* + \Sigma_\varepsilon))^{-1}(y - \iota^* A\widehat{h}_U), \qquad y \in K^*,$$

where

$$\widehat{h}_U = \hat{\mu} = \frac{1}{m} \sum_{j=1}^{m} x_j, \qquad \widehat{B}_U^2 = \widehat{\Sigma}_x = \frac{1}{m} \sum_{j=1}^{m} (x_j - \hat{\mu}) \otimes (x_j - \hat{\mu}).$$

Hence, in order to analyze the statistical properties of $R_{\widehat{h}_U, \widehat{B}_U}$, we first provide two concentration inequalities for $\widehat{\mu}$ and $\widehat{\Sigma}_x$, which are known since $x$ is a sub-Gaussian random vector in $X$. We include the proofs for the sake of completeness.

**Lemma A.10.** *Let $x$ be a $\kappa$ sub-Gaussian vector as in* (24). *Fix $\tau > 0$, then, with probability exceeding $1 - 2e^{-\tau}$,*

$$\|\widehat{\mu} - \mu\| \leq c\kappa \left( \sqrt{\frac{\mathrm{tr}(\Sigma_x)}{m}} + 2\sqrt{\frac{\tau \|\Sigma_x\|}{m}} \right), \tag{48}$$

*where $c > 0$ is a universal constant.*

*Proof.* Define $\xi = x - \mu$ where $\mu = \mathbb{E}[x]$. It is easy to show that $\xi$ is a zero mean $\kappa$ sub-Gaussian vector and its covariance matrix is $\Sigma_x = \mathbb{E}[\xi \otimes \xi]$. The assumption that $\xi$ is a sub-Gaussian random variable (24) can be equivalently expressed by requiring that

$$\|\langle \xi, v \rangle_X\|_{\psi_2} \leq \kappa \|\langle \xi, v \rangle_X\|_2,$$

where $\|\langle \xi, v \rangle_X\|_{\psi_2} = \sup_{p \geq 2} \frac{\|\langle \xi, v \rangle_X\|_p}{\sqrt{p}}$. For each $v$ in the unit ball $B_1$ of $X$, we set $\xi_v = \langle \xi, v \rangle_X$ and we regard $(\xi_v)_{v \in B_1}$ as a random process on $B_1$, viewed as metric space with respect to the metric $\mathrm{d}(v, w) = \|\xi_v - \xi_w\|_2$. Since $B_1 = -B_1$, then

$$\|\xi\|_X = \sup_{v \in B_1} |\langle \xi, v \rangle| = \sup_{v \in B_1} \langle \xi, v \rangle$$

and a standard result of random processes – see Exercise 8.6.5 and Theorem 8.5.5 in [48] – gives that

$$\mathbb{P}\left( \sup_{v \in B_1} |\xi_v| \leq c\kappa(W(B_1) + t\,\mathrm{diam}(B_1)) \right) \geq 1 - 2e^{-t^2}, \qquad t > 0,$$

where $c$ is a universal constant, $W$ is the width of the process

$$W(B_1) = \mathbb{E}[\sup_{v \in B_1} \langle \xi, v \rangle_X] = \mathbb{E}[\sup_{v \in B_1} |\langle \xi, v \rangle_X|] = \mathbb{E}[\|\xi\|_X],$$

and $\mathrm{diam}(B_1)$ is the diameter of $B_1$ with respect to the metric $\mathrm{d}(v, w)$

$$\mathrm{diam}(B_1) = \sup_{v, w \in B_1} \mathrm{d}(v, w) = \sup_{v, w \in B_1} \|\langle \xi, v \rangle - \langle \xi, w \rangle\|_2 = \sup_{v, w \in B_1} \mathbb{E}[\langle \xi, v - w \rangle^2]^{1/2}$$

$$= \sup_{v, w \in B_1} (\langle \Sigma_\xi (v - w), v - w \rangle)^{1/2} \leq 2\|\Sigma_\xi\|_{\mathcal{L}(X,X)}^{1/2}.$$

Hölder's inequality implies that

$$\mathbb{E}[\|\xi\|_X] \leq \mathbb{E}[\|\xi\|_X^2]^{1/2} = (\mathrm{tr}(\Sigma_x))^{1/2},$$

so that we get

$$\mathbb{P}\left( \|\xi\|_X \leq c\kappa \left( \sqrt{\mathrm{tr}\,\Sigma_x} + 2t\sqrt{\|\Sigma_x\|} \right) \right) \geq 1 - 2e^{-t^2}. \tag{49}$$

Define $\xi_1 = x_1 - \mu, \ldots, \xi_m = x_m - \mu$, which are i.i.d. as $\xi$. We claim that there exists an absolute constant $d$ such that $\widehat{\xi} = \frac{1}{m} \sum_{j=1}^{m} \xi_j$ is $d\kappa$ sub-Gaussian. Indeed,

$$\|\langle \sum_{j=1}^{m} \xi_j, v \rangle\|_{\psi_2}^2 = \|\sum_{j=1}^{m} \langle \xi_j, v \rangle\|_{\psi_2}^2$$

$$\leq d^2 \sum_{j=1}^{m} \|\langle \xi_j, v \rangle\|_{\psi_2}^2 \leq d^2 \kappa^2 \sum_{j=1}^{m} \|\langle \xi_j, v \rangle\|_2^2 = d^2 \kappa^2 \|\sum_{j=1}^{m} \langle \xi_j, v \rangle\|_2^2,$$

where the first inequality is due to the rotational invariance property of sub-Gaussian real random variables [48, Proposition 2.6.1], and the last equality is a consequence of the independence of the variables $\xi_j$. Thus $\widehat{\xi} = \widehat{\mu} - \mu$ is $d\kappa$ sub-Gaussian. Notice that the covariance of $\widehat{\xi}$ is given by

$$\mathbb{E}[\widehat{\xi} \otimes \widehat{\xi}] = \frac{1}{m^2} \sum_{i,j} \mathbb{E}[\xi_i \otimes \xi_j] = \frac{1}{m} \sum_j \mathbb{E}[\xi_j \otimes \xi_j] = \frac{1}{m} \Sigma_x.$$

Setting $\tau = t^2$, by applying (49) to $\widehat{\xi}$ we can finally deduce that

$$\mathbb{P}\left( \|\widehat{\xi}\|_X \leq cd\kappa \left( \sqrt{\frac{\operatorname{tr}\Sigma_x}{m}} + 2\sqrt{\frac{\tau\|\Sigma_x\|}{m}} \right) \right) \geq 1 - 2e^{-\tau},$$

which provides the claimed bound by redefining the universal constant $c$. $\qquad\square$

The following lemma is a restatement of a fundamental result in [30]. We include in the statement also the previous inequality.

**Lemma A.11.** *Let $x$ be a $\kappa$ sub-Gaussian vector as in* (24)*. Fix $\tau > 1$, then, with probability exceeding $1 - 3e^{-\tau}$,*

$$\|\widehat{\Sigma}_x - \Sigma_x\| \leq c\kappa^2 \|\Sigma_x\| \max\left\{ \sqrt{\frac{\operatorname{tr}\Sigma_x}{m\|\Sigma_x\|}}, \frac{\operatorname{tr}\Sigma_x}{m\|\Sigma_x\|}, \sqrt{\frac{\tau}{m}}, \frac{\tau}{m} \right\}, \tag{50}$$

$$\|\widehat{\mu} - \mu\| \leq c\kappa \left( \sqrt{\frac{\operatorname{tr}(\Sigma_x)}{m}} + 2\sqrt{\frac{\tau\|\Sigma_x\|}{m}} \right), \tag{51}$$

*where $c$ is a universal constant.*

*Proof.* We first introduce the operator

$$\widehat{\Sigma}_\xi = \frac{1}{m} \sum_{j=1}^m \xi_j \otimes \xi_j = \frac{1}{m} \sum_{j=1}^m (x_j - \mu) \otimes (x_j - \mu).$$

Since $\mathbb{E}[\widehat{\Sigma}_\xi] = \Sigma_x$, Theorem 9 of [30] gives that

$$\|\widehat{\Sigma}_\xi - \Sigma_x\| \leq c'\|\Sigma_x\| \max\left\{ \sqrt{\frac{\operatorname{tr}\Sigma_x}{m\|\Sigma_x\|}}, \frac{\operatorname{tr}\Sigma_x}{m\|\Sigma_x\|}, \sqrt{\frac{\tau}{m}}, \frac{\tau}{m} \right\}, \tag{52}$$

with probability greater than $1 - e^{-\tau}$. As usual, it holds that

$$\widehat{\Sigma}_x = \frac{1}{m} \sum_{j=1}^m (x_j - \widehat{\mu}) \otimes (x_j - \widehat{\mu}) = \frac{1}{m} \sum_{j=1}^m (x_j - \mu + \mu - \widehat{\mu}) \otimes (x_j - \mu + \mu - \widehat{\mu})$$

$$= \frac{1}{m} \sum_{j=1}^m (x_j - \mu) \otimes (x_j - \mu) + (\mu - \widehat{\mu}) \otimes \left( \frac{1}{m} \sum_{j=1}^m (x_j - \mu) \right)$$

$$+ \left( \frac{1}{m} \sum_{j=1}^m (x_j - \mu) \right) \otimes (\mu - \widehat{\mu}) + (\widehat{\mu} - \mu) \otimes (\widehat{\mu} - \mu)$$

$$= \widehat{\Sigma}_\xi - (\widehat{\mu} - \mu) \otimes (\widehat{\mu} - \mu).$$

As a consequence,

$$\|\widehat{\Sigma}_x - \Sigma_x\| \leq \|\widehat{\Sigma}_\xi - \Sigma_x\| + \|\widehat{\mu} - \mu\|^2.$$

By (52) and (48), with probability exceeding $1 - 3e^{-\tau}$, we have both (51) and

$$\|\widehat{\Sigma}_x - \Sigma_x\| \leq c'\|\Sigma_x\| \max\left\{ \sqrt{\frac{\operatorname{tr}\Sigma_x}{m\|\Sigma_x\|}}, \frac{\operatorname{tr}\Sigma_x}{m\|\Sigma_x\|}, \sqrt{\frac{\tau}{m}}, \frac{\tau}{m} \right\}$$

$$+ c\kappa^2 \left( \frac{\operatorname{tr}(\Sigma_x)}{m} + 4\frac{\sqrt{\tau\|\Sigma_x\|\operatorname{tr}(\Sigma_x)}}{m} + 4\frac{\tau\|\Sigma_x\|}{m} \right),$$

which provides the claimed bounds by redefining the constants $c$. $\qquad\square$

The following lemma shows that the excess risk $L(\widehat{h}_U, \widehat{B}_U) - L(h^\star, B^\star)$ is bounded by $\|\widehat{\Sigma}_x - \Sigma_x\|$ and $\|\widehat{\mu} - \mu\|$. Note that Lemma A.6 would provide a bound in terms of $\|\widehat{\Sigma}_x^{\frac{1}{2}} - \Sigma_x^{\frac{1}{2}}\|$. Since the square root is a monotone increasing function, it holds true that

$$\|\widehat{\Sigma}_x^{\frac{1}{2}} - \Sigma_x^{\frac{1}{2}}\| \leq \sqrt{\|\widehat{\Sigma}_x - \Sigma_x\|},$$

see Theorem X.1.1 of [5], which would provide a worse bound.

**Lemma A.12.** *Assume that $A\Sigma_x A^* + \Sigma_\varepsilon : Y \to Y$ has a bounded inverse, that the operator*

$$A^*(\iota^*(A\Sigma_x A^* + \Sigma_\varepsilon))^{-1} : \iota^*(Y) \subseteq K^* \to X$$

*extends to a bounded operator from $K^*$ to $X$, and that*

$$\|(A\Sigma_x A^* + \Sigma_\varepsilon)^{-1} A(\widehat{\Sigma}_x - \Sigma_x) A^*\| \leq 1/2. \tag{53}$$

*Then*

$$|L(\widehat{h}_U, \widehat{B}_U) - L(h^\star, B^\star)| = \mathrm{O}\left(\|\widehat{\Sigma}_x - \Sigma_x\|\right) + \mathrm{O}\left(\|\widehat{\mu} - \mu\|\right), \tag{54}$$

*where the constant in $\mathrm{O}$ only depends on $A, \Sigma_x, \Sigma_\varepsilon, \iota$ and $\mu$.*

*Proof.* Let

$$\widehat{x}_U = R_{\widehat{h}_U, \widehat{B}_U}(y), \qquad x^\star = R_{h^\star, B^\star}(y),$$

so that

$$L(\widehat{h}_U, \widehat{B}_U) - L(h^\star, B^\star) = \mathbb{E}\left[\|\widehat{x}_U - x\|^2\right] - \mathbb{E}\left[\|x^\star - x\|^2\right].$$

Since $x^\star$ minimizes the mean square error, clearly $L(\widehat{h}_U, \widehat{B}_U) - L(h^\star, B^\star) \geq 0$. We now prove the upper bound. Since

$$\|\widehat{x}_U - x\|^2 - \|x^\star - x\|^2 = \|\widehat{x}_U - x^\star\|^2 + 2\langle \widehat{x}_U - x^\star, x^\star - x\rangle$$
$$\leq \|\widehat{x}_U - x^\star\|^2 + 2\|\widehat{x}_U - x^\star\|\|x^\star - x\|,$$

then, by Hölder inequality,

$$\mathbb{E}\left[\|\widehat{x}_U - x\|^2\right] - \mathbb{E}\left[\|x^\star - x\|^2\right] \leq \mathbb{E}\left[\|x^\star - \widehat{x}_U\|^2\right] + 2\sqrt{\mathbb{E}\left[\|\widehat{x}_U - x^\star\|^2\right] \mathbb{E}\left[\|x^\star - x\|^2\right]}$$
$$= \mathrm{O}\left(\sqrt{\mathbb{E}\left[\|\widehat{x}_U - x^\star\|^2\right]}\right), \tag{55}$$

where the constant in $\mathrm{O}$ only depends on $A, \Sigma_x$ and $\Sigma_\varepsilon$. By (15) and the definition of $\widehat{x}_U$

$$x^\star = \Sigma_x A^*(\iota^*(A\Sigma_x A^* + \Sigma_\varepsilon))^{-1}(y - \iota^* A\mu) + \mu = \widetilde{W}y + \widetilde{b}$$
$$\widehat{x}_U = \widehat{\Sigma}_x A^*(\iota^*(A\widehat{\Sigma}_x A^* + \Sigma_\varepsilon))^{-1}(y - \iota^* A\widehat{\mu}) + \widehat{\mu} = \widehat{W}y + \widehat{b},$$

where $\widetilde{W}$ and $\widehat{W}$ are given in (33), for $B^2 = \Sigma_x$ and $B^2 = \widehat{\Sigma}_x$, respectively, and $\widetilde{b}, \widehat{b}$ in (35). As a consequence,

$$\widehat{x}_U - x^\star = (\widehat{W} - \widetilde{W})y + \widehat{b} - \widetilde{b} = Wy + b = W\iota^* A(x - \mu) + W\varepsilon + W\iota^* A\mu + b,$$

where

$$W = \widehat{W} - \widetilde{W}, \qquad b = \widehat{b} - \widetilde{b}.$$

Hence, taking into account that $x - \mu$ and $\varepsilon$ are zero mean random variables, we obtain

$$\mathbb{E}\left[\|\widehat{x}_U - x^\star\|^2\right]^{\frac{1}{2}} = \left(\mathrm{tr}\left[W(\iota^* A\Sigma_x A^*\iota + \iota^*\Sigma_\varepsilon \iota)W^*\right] + \|W\iota^* A\mu + b\|^2\right)^{\frac{1}{2}}$$
$$\leq \mathrm{tr}\left[W(\iota^* A\Sigma_x A^*\iota + \iota^*\Sigma_\varepsilon \iota)W^*\right]^{\frac{1}{2}} + \|W\iota^* A\mu + b\|$$
$$\leq \left(\mathrm{tr}\left(\iota^* A\Sigma_x A^*\iota + \iota^*\Sigma_\varepsilon \iota\right)\right)^{\frac{1}{2}} \|W\| + \|W\iota^* A\mu + b\|. \tag{56}$$

We now bound the norm of $W$ where

$$W = \widehat{\Sigma}_x A^*(\iota^*(A\widehat{\Sigma}_x A^* + \Sigma_\varepsilon))^{-1} - \Sigma_x A^*(\iota^*(A\Sigma_x A^* + \Sigma_\varepsilon))^{-1}.$$

A delicate issue is that $(\iota^*(A\widehat{\Sigma}_x A^* + \Sigma_\varepsilon))^{-1}$ and $(\iota^*(A\Sigma_x A^* + \Sigma_\varepsilon))^{-1}$ do not have bounded inverses, see the remark after Theorem 3.1. We first prove that $A\widehat{\Sigma}_x A^* + \Sigma_\varepsilon$ has a bounded inverse. Indeed, let $\Delta = A(\widehat{\Sigma}_x - \Sigma_x)A^*$, then

$$A\widehat{\Sigma}_x A^* + \Sigma_\varepsilon = (A\Sigma_x A^* + \Sigma_\varepsilon)\left(I + (A\Sigma_x A^* + \Sigma_\varepsilon)^{-1}\Delta\right).$$

By assumption (53), $\|(A\Sigma_x A^* + \Sigma_\varepsilon)^{-1}\Delta\| \leq 1/2 < 1$, so that using Neumann series we have that $A\widehat{\Sigma}_x A^* + \Sigma_\varepsilon$ is invertible and

$$\begin{aligned}
&(A\widehat{\Sigma}_x A^* + \Sigma_\varepsilon)^{-1} - (A\Sigma_x A^* + \Sigma_\varepsilon)^{-1}\\
&= \left(I + (A\Sigma_x A^* + \Sigma_\varepsilon)^{-1}\Delta\right)^{-1}(A\Sigma_x A^* + \Sigma_\varepsilon)^{-1} - (A\Sigma_x A^* + \Sigma_\varepsilon)^{-1}\\
&= \left(\left(I + (A\Sigma_x A^* + \Sigma_\varepsilon)^{-1}\Delta\right)^{-1} - I\right)(A\Sigma_x A^* + \Sigma_\varepsilon)^{-1}\\
&= \left(I + (A\Sigma_x A^* + \Sigma_\varepsilon)^{-1}\Delta\right)^{-1}(A\Sigma_x A^* + \Sigma_\varepsilon)^{-1}\Delta(A\Sigma_x A^* + \Sigma_\varepsilon)^{-1}.
\end{aligned}$$

Then, on $\iota^*(Y) \subseteq K^*$

$$\begin{aligned}
&(i^*(A\widehat{\Sigma}_x A^* + \Sigma_\varepsilon))^{-1} - (i^*(A\Sigma_x A^* + \Sigma_\varepsilon))^{-1}\\
&= \left(I + (A\Sigma_x A^* + \Sigma_\varepsilon)^{-1}\Delta\right)^{-1}(A\Sigma_x A^* + \Sigma_\varepsilon)^{-1}A(\widehat{\Sigma}_x - \Sigma_x)A^*(i^*(A\Sigma_x A^* + \Sigma_\varepsilon))^{-1}.
\end{aligned}$$

The density of $\iota^*(Y) \subset K^*$ and the assumption that $A^*(i^*(A\Sigma_x A^* + \Sigma_\varepsilon))^{-1}$ extends to a bounded operator from $K^*$ to $X$ implies that

$$\begin{aligned}
&\|(i^*(A\widehat{\Sigma}_x A^* + \Sigma_\varepsilon))^{-1} - (i^*(A\Sigma_x A^* + \Sigma_\varepsilon))^{-1}\|\\
&\leq \|\left(I + (A\Sigma_x A^* + \Sigma_\varepsilon)^{-1}\Delta\right)^{-1}\|\|(A\Sigma_x A^* + \Sigma_\varepsilon)^{-1}A\|\|\widehat{\Sigma}_x - \Sigma_x\|\|A^*(i^*(A\Sigma_x A^* + \Sigma_\varepsilon))^{-1}\|\\
&\leq 2\|(A\Sigma_x A^* + \Sigma_\varepsilon)^{-1}A\|\|\widehat{\Sigma}_x - \Sigma_x\|\|A^*(i^*(A\Sigma_x A^* + \Sigma_\varepsilon))^{-1}\|,
\end{aligned}$$

where we used (53) to bound $\|\left(I + (A\Sigma_x A^* + \Sigma_\varepsilon)^{-1}\Delta\right)^{-1}\|$ with 2, so that

$$\|(i^*(A\widehat{\Sigma}_x A^* + \Sigma_\varepsilon))^{-1} - (i^*(A\Sigma_x A^* + \Sigma_\varepsilon))^{-1}\| \lesssim \|\widehat{\Sigma}_x - \Sigma_x\|, \tag{57}$$

where the constant in $\lesssim$ only depends on $A, \Sigma_x$ and $\Sigma_\varepsilon$. Since

$$\begin{aligned}
W &= \widehat{\Sigma}_x A^*\left((\iota^*(A\widehat{\Sigma}_x A^* + \Sigma_\varepsilon))^{-1} - (\iota^*(A\Sigma_x A^* + \Sigma_\varepsilon))^{-1}\right)\\
&\quad + (\widehat{\Sigma}_x - \Sigma_x)A^*(\iota^*(A\Sigma_x A^* + \Sigma_\varepsilon))^{-1},
\end{aligned}$$

eq. (57) and the fact that $\|\widehat{\Sigma}_x\| \leq \|\widehat{\Sigma}_x - \Sigma_x\| + \|\Sigma_x\|$ both imply

$$\|W\| \lesssim \|\widehat{\Sigma}_x - \Sigma_x\| + \|\widehat{\Sigma}_x - \Sigma_x\|^2 = O\left(\|\widehat{\Sigma}_x - \Sigma_x\|\right), \tag{58}$$

where the constants in $\lesssim$ and O only depend on $A, \Sigma_x$ and $\Sigma_\varepsilon$. We now observe that

$$b = (\widehat{\mu} - \mu) - (\widehat{W}\iota^* A\widehat{\mu} - \widetilde{W}\iota^* A\mu) = (\widehat{\mu} - \mu) - \widehat{W}\iota^* A(\widehat{\mu} - \mu) - W\iota^* A\mu,$$

so that

$$W\iota^* A\mu + b = (I - \widehat{W}\iota^* A)(\widehat{\mu} - \mu),$$

and

$$\|W\iota^* A\mu + b\| \leq \|(I - \widehat{W}\iota^* A)\|\|\widehat{\mu} - \mu\| \leq (\|(I - \widetilde{W}\iota^* A)\| + \|W\iota^* A\|)\|\widehat{\mu} - \mu\|.$$

Eq. (58) implies that

$$\|W\iota^* A\mu + b\| \lesssim \|\widehat{\mu} - \mu\| + \|\widehat{\mu} - \mu\|\|\widehat{\Sigma}_x - \Sigma_x\| = O(\|\widehat{\mu} - \mu\|), \tag{59}$$

where the constants in $\lesssim$ and O only depend on $A, \Sigma_x, \Sigma_\varepsilon$ and $\mu$. Eqs. (55) and (56) with (59) give (54). $\qquad\square$

We are now able to prove the main result of section 4.2.

*Proof of Theorem 4.2 .* Since the map $C \mapsto (A\Sigma_x A^* + \Sigma_\varepsilon)^{-1} ACA^*$ is continuous from $\mathcal{L}(X,X)$ into $\mathcal{L}(Y,Y)$, there exists $\delta > 0$ such that

$$\|(A\Sigma_x A^* + \Sigma_\varepsilon)^{-1} ACA^*\| \le 1/2 \qquad \forall C \in \mathcal{L}(X,X) \quad \|C\| \le \delta.$$

Set $m_0 \in \mathbb{N}$ such that

$$c\kappa^2 \|\Sigma_x\| \max\left\{ \sqrt{\frac{\operatorname{tr}\Sigma_x}{m_0 \|\Sigma_x\|}}, \frac{\operatorname{tr}\Sigma_x}{m_0 \|\Sigma_x\|}, \sqrt{\frac{\tau}{m_0}}, \frac{\tau}{m_0} \right\} \le \delta,$$

where $c$ is the constant in Lemma A.11. Eq. (50) implies that for all $m \ge m_0$ condition (53) is satisfies with probability exceeding $1 - 4e^{-\tau}$. Possibly redefining $m_0$, by (50) and (51) we can assume that on the same event

$$\max\{\|\widehat{\mu} - \mu\|, \|\widehat{\Sigma_x} - \Sigma_x\|\} \le \min\{1, \frac{c_1 + c_2\sqrt{\tau}}{\sqrt{m}}\}, \tag{60}$$

where $c_1$ and $c_2$ are suitable constants independent of $m$ and $\tau$. Hence, eq. (54) implies that on the same event

$$|L(\widehat{h}_U, \widehat{B}_U) - L(h^\star, B^\star)| = \mathrm{O}\left(\|\widehat{\Sigma_x} - \Sigma_x\|\right) + \mathrm{O}\left(\|\widehat{\mu} - \mu\|\right) \le C \frac{c_1 + c_2\tau}{\sqrt{m}},$$

where the last inequality is a consequence of (60). Eq. (25) is now clear. $\square$

## A.7 Numerical results: further details

### A.7.1 Experimental setup

In Section 5, we set $X = L^2(\mathbb{T}^1)$, being $\mathbb{T}^1$ the one-dimensional torus. For any $N > 0$, we can introduce the partition $\{I_{N,i}\}_{i=1}^N$ of the interval $(0,1)$, being $I_{N,i} = \left(\dfrac{i-1}{N}, \dfrac{i}{N}\right)$ and define the 1D-pixel basis $\{\varphi_{N,i}\}_{i=1}^N$ as follows:

$$\varphi_{N,i}(t) = \sqrt{N}\chi_{N,i}(t), \qquad \chi_{N,i}(t) = \begin{cases} 1 & t \in I_{N,i}, \\ 0 & \text{otherwise.} \end{cases}$$

The functions $\{\varphi_{N,i}\}_{i=1}^N$ form an orthogonal set, and we define $X_N$ as the linear space generated by them. Each element $u \in X_N$ can be uniquely represented by a vector $\boldsymbol{u} \in \mathbb{R}^N$ as follows:

$$\boldsymbol{u}_i = \frac{1}{|I_{N,i}|} \int_{I_{N,i}} u = \sqrt{N}\langle u, \varphi_{N,i}\rangle_X, \qquad u(t) = \sum_{i=1}^N \langle u, \varphi_{N,i}\rangle_X \varphi_{N,i}(t) = \sum_{i=1}^N \frac{1}{\sqrt{N}} \boldsymbol{u}_i \varphi_{N,i}(t).$$

As a consequence, for $u \in X_N$, we can compute $\|u\|_X^2 = \sum_{i=1}^N \langle u, \varphi_{N,i}\rangle^2 = \frac{1}{N}\sum_{i=1}^N \boldsymbol{u}_i^2$. The representation of a linear operator $B: X_N \to X_N$ can be done via a matrix $\boldsymbol{B} \in \mathbb{R}^{N\times N}$ as follows:

$$\boldsymbol{B}_{i,j} = \langle B\varphi_{N,j}, \varphi_{N,i}\rangle_X; \qquad v = Bu \iff \boldsymbol{v} = \boldsymbol{B}\boldsymbol{u}.$$

In order to generate a discrete version of the random process $\varepsilon$ and of the random variable $x$, we first generate the vectors $\boldsymbol{\nu}_x, \boldsymbol{\nu}_\varepsilon$ such that each component $[\boldsymbol{\nu}_x]_i$ and $[\boldsymbol{\nu}_\varepsilon]_i$ is independently distributed with mean 0 and covariance 1. In the proposed tests, we either draw from a Gaussian distribution $\mathcal{N}(0,1)$ or a uniform distribution $\mathrm{Unif}(-\sqrt{3}, \sqrt{3})$, taking advantage of the Matlab commands `randn` and `rand`. Then in order to approximate the white noise process $\varepsilon$, with zero mean and covariance operator $\Sigma_\varepsilon = \sigma^2 I$, we introduce $\boldsymbol{\varepsilon}$ such that

$$\mathbb{E}[\boldsymbol{\varepsilon}_i \boldsymbol{\varepsilon}_j] = \mathbb{E}[\sqrt{N}\langle \varepsilon, \varphi_{N,i}\rangle \sqrt{N}\langle \varepsilon, \varphi_{N,j}\rangle] = \sigma^2 N \delta_{ij},$$

thus resulting in $\boldsymbol{\varepsilon} = \sigma\sqrt{N}\boldsymbol{\nu}_\varepsilon$. As an alternative, we also consider a random process whose components with respect to the Haar wavelet basis are randomly sampled as a white noise, i.e., $\boldsymbol{\varepsilon} = \sigma\sqrt{N}\boldsymbol{W}^T\boldsymbol{\nu}_\varepsilon$, where $\boldsymbol{W}$ is the discrete Haar wavelet transform and $\boldsymbol{W}^T$ its transpose.

The random variable $\boldsymbol{x}$ is instead computed as $\boldsymbol{x} = \boldsymbol{\mu} + \sqrt{N}\boldsymbol{\Sigma}_x^{1/2}\boldsymbol{\nu}_x$, being

$$\boldsymbol{\mu}_i = \sqrt{N}\langle \mu, \varphi_{N,i}\rangle_X, \qquad [\boldsymbol{\Sigma}_x^{1/2}]_{i,j} = \langle \Sigma_x^{1/2}\varphi_{N,j}, \varphi_{N,i}\rangle_X.$$

In the experiments, we picked $\mu(t) = 1 - |2t - 1|$ and $\Sigma_x^{1/2}$ s.t.

$$\Sigma_x^{1/2} u(t) = \int_{\mathbb{T}^1} k_{\Sigma_x}(t') u(t - t') dt, \quad k_{\Sigma_x}(t) = 1 - \exp(-(c/t)^4)\chi_{(-c,c)}(t),$$

being $c = 0.2$. Finally, we selected $\sigma = 0.05$. In Figure 3, we show some signals from the training sample, both in dimension $N = 64$ and $N = 256$.

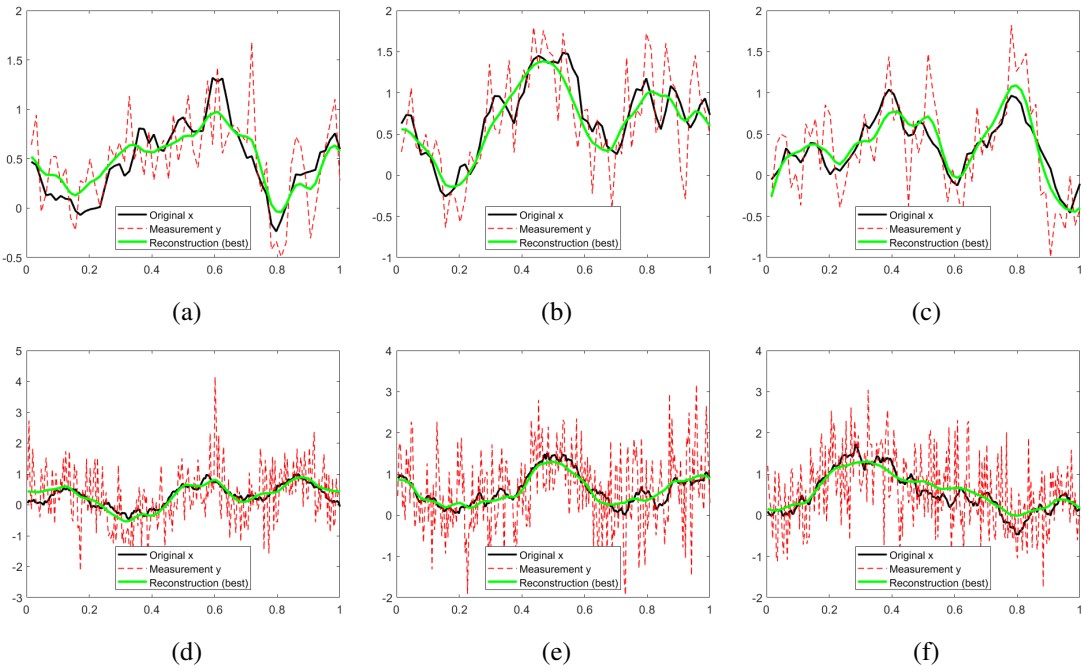

Figure 3: Signals drawn from the joint distribution in the case where both $x$ and $\varepsilon$ are Gaussian. (a),(b),(c): $N = 64$, (d),(e),(f): $N = 256$. We show in black the original signal $x$, in red the noisy datum $y = x + \varepsilon$ and in green the reconstruction $R_{\theta^\star}(y)$ associated with the optimal regularizer.

### A.7.2 Implementation aspects

As expressed in Section 5, it is possible to compute the mean squared error $L$ associated with the optimal parameter $\theta^\star = (h^\star, B^\star)$ and the learned parameters $\widehat{\theta}_S = (\widehat{h}_S, \widehat{B}_S)$, $\widehat{\theta}_U = (\widehat{h}_U, \widehat{B}_U)$ with an explicit formula. Indeed, since the employed data are synthetically generated, we can take advantage of the knowledge of $\mu, \Sigma_x, \Sigma_\varepsilon$. We simulate the computations in Section 3 and in Appendix A.3, in a finite-dimensional context: here, since for any $N$ the operator $\Sigma_\varepsilon$ is invertible, Assumption 2.8 is satisfied with $K = Y$. The expression of the regularizer in (12) then reads as

$$R_{h,B}(y) = Wy + b,$$

being $W = B^2 A^* (\Sigma_\varepsilon + A B^2 A^*)^{-1}$ and $b = (I_X - WA)h$. Moreover, as in Appendix A.3, we can compute

$$L(h, B) = \mathrm{tr}[(WA - I_X)\Sigma_x(WA - I_X)^*] + \mathrm{tr}[W\Sigma_\varepsilon W^*] + \|(WA - I_X)\mu + b\|_X^2.$$

By this formula, it is possible to compute the mean squared error associated to any parameter $\theta$, and in particular for $(h^\star, B^\star) = (\mu, \Sigma_x^{1/2})$ and $(\widehat{h}_U, \widehat{B}_U) = (\widehat{\mu}, \widehat{\Sigma}_x^{1/2})$.

In order to detect the empirical risk minimizer $(\widehat{h}_S, \widehat{B}_S)$, and in particular to compute the quantity $L(\widehat{h}_S, \widehat{B}_S)$, we rely on the same strategy adopted for the minimization of $L$. Therefore, we first look for the affine functional $Wy + b$ which minimizes the empirical risk, defined as

$$\widehat{L}_{b,W} = \frac{1}{m} \sum_{j=1}^{m} \|Wy_j + b - x_j\|_X^2;$$

then, if the optimal $b$ and $W$ can be written as $W = B^2 A^* (\Sigma_\varepsilon + A B^2 A^*)^{-1}$ and $b = (I_X - WA)h$, the pair $(h, B)$ is a minimizer of $\widehat{L}(h, B)$. Thanks to the empirical mean and covariance matrices

$$\widehat{y} = \frac{1}{m} \sum_{j=1}^{m} y_j, \qquad \widehat{\Sigma}_y = \frac{1}{m} \sum_{j=1}^{m} (y_j - \widehat{y}) \otimes (y_j - \widehat{y}), \qquad \widehat{\Sigma}_{yx} = \frac{1}{m} \sum_{j=1}^{m} (y_j - \widehat{y}) \otimes (x_j - \widehat{\mu}),$$

it is also possible to provide a more explicit formula for $\widehat{L}_{b,W}$. Indeed,

$$\widehat{L}_{b,W} = \frac{1}{m}\sum_{j=1}^{m}\left(\|W(y_j - \widehat{y})\|_X^2 + \|x_j - \widehat{\mu}\|_X^2 - 2\langle W(y_j - \widehat{y}), x_j - \widehat{\mu}\rangle_X + \|W\widehat{y} - \widehat{\mu} + b\|_X^2\right)$$

$$= \mathrm{tr}[W\widehat{\Sigma}_y W^*] + \mathrm{tr}[\widehat{\Sigma}_x] - 2\,\mathrm{tr}[W\widehat{\Sigma}_{yx}] + \|W\widehat{y} - \widehat{\mu} + b\|_X^2,$$

where we have used that $\sum_j(y_j - \widehat{y}) = 0$ and $\sum_j(x_j - \widehat{\mu}) = 0$. Thus, the minimizer of $\widehat{L}_{b,W}$ is the affine operator associated with $W = \widehat{\Sigma}_{xy}\widehat{\Sigma}_y^{-1}$ and $b = \widehat{\mu} - W\widehat{y}$. Unfortunately, such $W$ does not yield the optimal parameter $\widehat{B}_S$: indeed, $W$ cannot be written in the form $W = B^2 A^*(\Sigma_\varepsilon + AB^2A^*)^{-1}$, but rather $W = MA^*(\Sigma_\varepsilon + AMA^*)^{-1}$, where the resulting $M$ is not symmetric. We overcome such issue by considering the symmetric part of $M$, which we denote by $M'$. Indeed, despite the operator $W'$ associated with $M'$ is possibly different from the minimizer of $\widehat{L}_{b,W}$ among the functionals of the form $W = B^2 A^*(\Sigma_\varepsilon + AB^2A^*)^{-1}$, numerical evidence shows that the values of $L$ evaluated in $W'$ and $W$ are very close. Since the former is an upper bound of $L(\widehat{h}_S, \widehat{B}_S)$ and the latter a lower bound, we conclude that the expected loss $L$ evaluated in $W'$ provides a sufficiently tight upper estimate of the value of $L(\widehat{h}_S, \widehat{B}_S)$, without explicitly requiring the computation of $\widehat{B}_S$ and $\widehat{h}_S$.

As a final remark, we show how the generalization bounds in probability obtained in Theorems 4.1 and 4.2 can be reformulated in expectation. Let us first consider the unsupervised case, in which

$$\mathbb{P}_{\mathbf{z}\sim\rho^m}\left[|L(\widehat{h}_U, \widehat{B}_U) - L(h^\star, B^\star)| \le c_3\frac{1}{\sqrt{m}} + c_4\frac{\sqrt{\tau}}{\sqrt{m}}\right] \ge 1 - e^{-\tau}.$$

Inverting $\eta = c_3\frac{1}{\sqrt{m}} + c_4\frac{\sqrt{\tau}}{\sqrt{m}}$ in terms of $\tau$ we get

$$\mathbb{P}_{\mathbf{z}\sim\rho^m}\left[|L(\widehat{h}_U, \widehat{B}_U) - L(h^\star, B^\star)| \le \eta\right] \ge 1 - \widetilde{c}_1 e^{-\widetilde{c}_2 m\eta^2}.$$

This can be translated into a bound in expectation by means of the following identity:

$$\mathbb{E}_{\mathbf{z}\sim\rho^m}\left[|L(\widehat{h}_U, \widehat{B}_U) - L(h^\star, B^\star)|\right] = \int_0^\infty \mathbb{P}_{\mathbf{z}\sim\rho^m}\left[|L(\widehat{h}_U, \widehat{B}_U) - L(h^\star, B^\star)| > \eta\right]d\eta \lesssim \frac{1}{\sqrt{m}}.$$

In a similar way, in the supervised case we have (see the Appendix A.5)

$$\mathbb{P}_{\mathbf{z}\sim\rho^m}\left[|L(\widehat{\theta}_{\mathbf{z}}) - L(\theta^*)| \le \eta\right] \ge 1 - e^{c_1\eta^{-1/s'} - c_2 m\eta^2}.$$

Notice that, when $\eta \to 0$, such bound could be meaningless, as the term $e^{c_1\eta^{-1/s'}}$ blows up. We therefore substitute it with the following estimate:

$$\mathbb{P}_{\mathbf{z}\sim\rho^m}\left[|L(\widehat{\theta}_{\mathbf{z}}) - L(\theta^*)| \le \eta\right] \ge 1 - \min\{1, e^{c_1\eta^{-1/s'} - c_2 m\eta^2}\}.$$

As a consequence,

$$\mathbb{E}_{\mathbf{z}\sim\rho^m}\left[|L(\widehat{h}_U, \widehat{B}_U) - L(h^\star, B^\star)|\right] = \int_0^\infty \mathbb{P}_{\mathbf{z}\sim\rho^m}\left[|L(\widehat{h}_U, \widehat{B}_U) - L(h^\star, B^\star)| > \eta\right]d\eta$$

$$\le \int_0^\infty \min\{1, e^{c_1\eta^{-1/s'} - c_2 m\eta^2}\}d\eta.$$

Notice that $1 \le e^{c_1\eta^{-1/s'} - c_2 m\eta^2}$ when $c_1\eta^{-1/s'} \ge c_2 m\eta^2$, namely when $\eta \le \widehat{\eta}(m) = \left(\frac{c_1}{c_2 m}\right)^{\frac{1}{2+1/s'}}$. Thus,

$$\mathbb{E}_{\mathbf{z}\sim\rho^m}\left[|L(\widehat{h}_U, \widehat{B}_U) - L(h^\star, B^\star)|\right] \le \widehat{\eta}(m) + \int_{\widehat{\eta}(m)}^\infty e^{c_1\eta^{-1/s'} - c_2 m\eta^2}d\eta$$

$$\le \widehat{\eta}(m) + \int_{\widehat{\eta}}^\infty e^{c_1\widehat{\eta}^{-1/s'} - c_2 m\eta^2}d\eta = \left[\begin{array}{l} c_1\widehat{\eta}^{-1/s'} - c_2 m\eta^2 = -\beta^2 \\ d\eta = \frac{1}{\sqrt{c_2 m}}\frac{\beta}{\sqrt{c_1\widehat{\eta}^{-1/s'} + \beta^2}}d\beta \le \frac{1}{\sqrt{c_2 m}}d\beta \end{array}\right]$$

$$\le \widehat{\eta}(m) + \frac{1}{\sqrt{c_2 m}}\int_0^\infty e^{-\beta^2}d\beta \lesssim \left(\frac{1}{m}\right)^{\frac{1}{2+1/s'}} + \frac{1}{\sqrt{m}},$$

| | Sample size, $m$ | | | | | | |
|---|---|---|---|---|---|---|---|
| Model (1) | 3000 | 6463 | 13925 | 30000 | 64633 | 139248 | 300000 |
| $N = 64$, unsup. | 0.00174 | 0.00119 | 0.00076 | 0.00056 | 0.00038 | 0.00025 | 0.00018 |
| $N = 64$, sup. | 0.00398 | 0.00233 | 0.00156 | 0.00111 | 0.00067 | 0.00044 | 0.00031 |
| $N = 256$, unsup. | 0.00195 | 0.00131 | 0.00086 | 0.00063 | 0.00040 | 0.00029 | 0.00020 |
| $N = 256$, sup. | 0.01369 | 0.00485 | 0.00246 | 0.00134 | 0.00092 | 0.00052 | 0.00037 |
| Model (2) | | | | | | | |
| $N = 64$, unsup. | 0.00177 | 0.00129 | 0.00082 | 0.00053 | 0.00034 | 0.00023 | 0.00019 |
| $N = 64$, sup. | 0.00380 | 0.00236 | 0.00158 | 0.00103 | 0.00058 | 0.00044 | 0.00032 |
| $N = 256$, unsup. | 0.00199 | 0.00126 | 0.00094 | 0.00056 | 0.00044 | 0.00028 | 0.00019 |
| $N = 256$, sup. | 0.01449 | 0.00487 | 0.00250 | 0.00142 | 0.00086 | 0.00058 | 0.00035 |
| Model (3) | | | | | | | |
| $N = 64$, unsup. | 0.00188 | 0.00125 | 0.00083 | 0.00056 | 0.00039 | 0.00027 | 0.00018 |
| $N = 64$, sup. | 0.00407 | 0.00240 | 0.00154 | 0.00105 | 0.00069 | 0.00044 | 0.00028 |
| $N = 256$, unsup. | 0.00190 | 0.00134 | 0.00088 | 0.00057 | 0.00040 | 0.00028 | 0.00019 |
| $N = 256$, sup. | 0.01434 | 0.00503 | 0.00248 | 0.00135 | 0.00089 | 0.00052 | 0.00036 |

Table 1: Tabulated values of the excess risks associated with Figures 1(c),**??**(c),**??**(c), computed at two discretization levels and in three different statistical setups: Gaussian variable $x$ and (1) uniform white noise $\varepsilon$, (2) Gaussian white noise $\varepsilon$, and (3) white noise $\varepsilon$ uniformly distributed w.r.t. the Haar wavelet transform.

and the leading order is $\left(\frac{1}{m}\right)^{\frac{1}{2+1/s'}}$, which can be rewritten as $\left(\frac{1}{\sqrt{m}}\right)^{1-\frac{1}{2s'+1}}$, and converges to $\frac{1}{\sqrt{m}}$ for large values of $s'$ (namely, of $s$).

Finally, in Table 1 we report the numerical values of the excess risk $|L(\widehat{h}_U, \widehat{B}_U) - L(h^\star, B^\star)|$ and $|L(\widehat{h}_S, \widehat{B}_S) - L(h^\star, B^\star)|$ associated with all the studied cases.

## A.8 An ill-posed inverse problem: deconvolution of 1D signals

We provide a numerical verification of the estimates of Theorems 4.1 and 4.2 for a 1D deconvolution problem, extending the experiments of section 5 to the case of an ill-posed operator $A$. We consider again $X = Y = L^2(\mathbb{T}^1)$, and introduce the convolution operator $(Ax)(t) = (k * x)(t) = \int_{\mathbb{T}} k(t - \tau)x(\tau)d\tau$. This operation can be used to describe the blurring of one-dimensional signals, the function $k$ being the convolutional filter, or point spread function. In our experiments, we consider $k(t) = \chi_{[-L,L]}(t)$, the indicator function of the interval $[-L, L]$, and set $L = 0.02$. Such a kernel $k$ can be referred to as the average filter. When discretizing the interval $\mathbb{T}^1$ with $N$ 1D-pixels, the operator $A$ reduces to a discrete (periodic) convolution with a constant vector $\mathbf{k}$, whose number of entries is $LN$. Both at a continuous and at a discrete level, the deconvolution problem is known to be ill-posed (see, e.g., [38]), and the smallest singular value of the discretized operator vanishes as $N$ grows. Nevertheless, we expect to observe the same generalization bounds as in the denoising case.

We replicate the same experiments as in section 5, assuming that $x$ is a random Gaussian variable (with mean $\mu$ and covariance $\Sigma_x$ as reported in section 5) and $\varepsilon$ is white uniform noise with covariance $\Sigma_\varepsilon = \sigma^2 I$. We fix a noise level of 2.5% by setting $\sigma$ equal to the 2.5% of the peak value of the average signal. The results of the numerical experiments are reported in Figure 4. We observe that in both scenarios the decay of the excess risk is of the order $1/\sqrt{m}$, and the unsupervised technique still provides (slightly) better results, which in particular are not affected by the increased ill-posedness of the operator at a much refined scale.

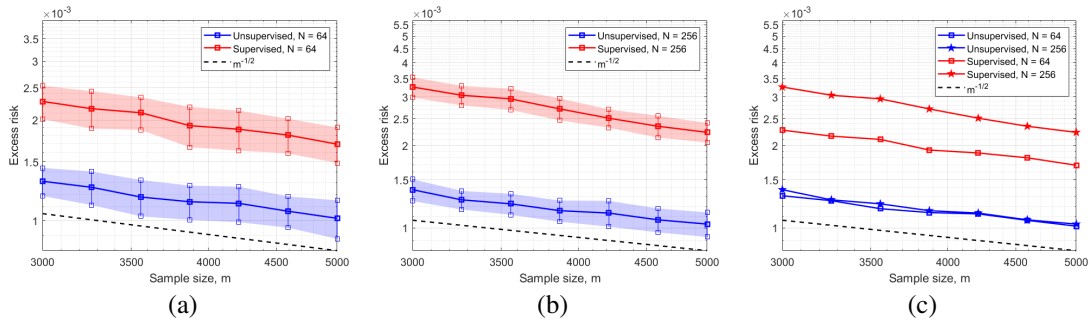

Figure 4: Decay of the excess risks $|L(\widehat{\theta}_S) - L(\theta^\star)|$ and $|L(\widehat{\theta}_U) - L(\theta^\star)|$ (with standard deviation error bars) with two different discretization sizes, $N = 64$ (a) and $N = 256$ (b), and comparison (c).