# OpenReview forum: "Learning the optimal Tikhonov regularizer for inverse problems"
_NeurIPS.cc/2021/Conference — NeurIPS 2021 Poster_

### Official Review · Reviewer_N3Qr · 2021-07-05

**Rating:** 6
**Confidence:** 3

**Summary:**

This paper considers solving the inverse problem with a quadratic regularizer. It shows that the optimal parameters for the regularizer, in terms of achieving the smallest reconstruction error, are given by the mean and covariance of the true signal distribution. The paper also provides upper bounds for excess risk (expected error) for the estimated parameters learned from a training data set in either a supervised or unsupervised manner.

**Limitations And Societal Impact:**

I don't think there is a limitation or potential negative societal impact of this work.

**Main Review:**

Strength
This paper considers solving the inverse problem with a quadratic regularizer (generalized Tikhonov regularization). The problem is formulated rigorously in the context of functionals. There are two parts of theoretical results provided by this paper.  Firstly, it shows that the optimal shifting and scaling parameters in the regularizer should be the mean and covariance of the true signal distribution. Secondly, when the distribution is not available, the paper suggests two ways to learn those parameters from data and provides upper bounds for the excess risk. The results are verified through experiments.

Weakness
1. The paper shows the optimal parameters for the regularizer are the mean and covariance. By plugging them into the objective, it reduces to the objective of MAP, up to a scalar factor. It would be better to discuss this connection in the paper.
2. Theorem 4.1 relies on the assumption (a), which introduces another Hilbert space H and corresponding maps j1 and j2. It seems unnatural and abstract to me to introduce those notations even for the purpose of bounding covering numbers. It would be better to discuss how this assumption fits into the practice.
3. It is interesting that the unsupervised method outperforms the supervised one in terms of both theory and simulation. Considering that the supervised method requires additional data of y and more computation, the value of using the supervised method is further decreased. The information from y or the error seems unnecessary in estimating h and B, while the paper doesn't provide any discussion on this.
4. In the numerical analysis, the paper only considers a simple case where A=I. That seems not enough since A is really an important component of the inverse problem. The experiments with a more general A are necessary and helpful in justifying the theoretical results.


---- after rebuttal ----

I appreciate the author's feedback. After reading the response and other reviews, I would like to keep my rating for this paper.

**Time Spent Reviewing:**

4

---

> ### Author Response · Authors · 2021-08-09
> **Reply to Official Review of Paper10535 by Reviewer N3Qr**
>
> We would like to thank the reviewer for the positive feedback and the useful comments and suggestions. We address all of them below.
>
> 1. This is indeed an important point, we will mention this connection in the remarks after Theorem 3.1. We remark that, in a Bayesian framework, the interpretation of the optimal Tikhonov regularizer as a MAP estimator is only valid for Gaussian random variables, whereas the result we propose in Section 3 is valid for a larger class of random variables and processes. We propose to add also the references [Gribonval, IEEE Trans. Signal Process. 2011] and [Gribonval-Machart, NIPS 2013], which discuss the connection of MAP estimation and non-quadratic variational regularization.
> 2. We propose to add the following example about assumption a) at the root of  Theorem 4.1. When $H$ and $X$ are Sobolev spaces of periodic functions, $H=H^{\sigma_1}(\mathbb{T}^1)$ and $X=H^{\sigma_2}(\mathbb{T}^1)$ the assumption that $j_1:H\to X$ satisfies assumption a) is just that $\sigma_1>\sigma_2$. Thus, assumption a) is typically satisfied in most imaging applications. In rough terms, it is analogous to assuming that the measured signal is smoother than the white noise measurement error. A similar argument applies to $B^\star$. It is worth observing that smoothness assumptions are very common in machine learning in order to obtain explicit bounds, see  [Blanchard-Mücke, FoCM, 2018] and references therein. We will add a sentence explaining the meaning of such assumption, and add the quoted reference.
> 3. This is a very relevant consideration that can be derived from our work. It was already briefly mentioned at the beginning of Section 4.2, but without referring to the (not yet available) generalization bounds.  We will certainly extend this discussion and add a brief comment on this aspect in the conclusions.
> 4. We implemented also the case of ill-posed operators, and in particular of operators whose singular values decay polynomially. In the outcome of the experiments, we observe the same decay of the excess risk as in the denoising case, up to different constants. We propose to include the related numerical results and a comment in a new Appendix A.8.

---

### Official Review · Reviewer_NW21 · 2021-07-12

**Rating:** 7
**Confidence:** 3

**Summary:**

The paper studies the linear MMSE in infinite-dimensional systems. Specifically, given an observation $Ax +\varepsilon$ where $A$ is a known linear operator and $\varepsilon$ is some form of noise, one wants to estimate $x$, which minimizes the mean squared error.

Focusing on regularization, the paper asks what is the optimal regularizer with respect to the mean squared error. It is shown that, when constrained to Tikhonov regularization, the optimal regularizer has a rather simple form which depends only on the statistical properties of the system and, in particular, is independent from the operator $A$.

The authors also address the problem of approximating the regularizer, based on a finite sample. They show two approximations schemes: A "supervised-learning" scheme, based on minimizing the empirical risk and an "unsupervised-learning" scheme, which is based on the particular form the optimal regularizer was found to have.

For both schemes, the authors show that as the sample size $m$ tend to infinity, the approximation error decays at the expected rate $\frac{1}{\sqrt{m}}$.

**Limitations And Societal Impact:**

The authors adequately addressed the limitations and potential negative societal impact of their work.

**Main Review:**

The main results of the paper are a generalization of well known results in finite dimensions to an infinite-dimensional settings.
The main ideas which underlies the proofs are also derived from the finite dimensional setting.
E.g, in order to find the LMMSE, the authors equate the Gateaux derivative to 0, instead of the 'regular' derivative, as is done in the classical proofs.
In this respect, the results are not surprising and perhaps, one could claim, are not very exciting.

However, as the authors demonstrate, the move to infinite-dimensions has many technical difficulties and is a highly non-trivial task.
Special care must be taken when discussing the type of spaces involved in the problem, the type of linear operator between those spaces and even the form of admissible randomness. Indeed, even the correct formulation of the problem is not immediate, since in general one should not expect $A$ to have a bounded inverse.

The paper is well written, and the authors do a rather good job in motivating the formulation of the problem and the definitions. The math itself is very elegant and it could even prove useful for other, similar, problems. I think it can fit well with the theme of conference.

Having said the above, I do feel that the style and presentation of the paper could be improved. I add some concrete suggestions/comments below:

- Above all, I feel that the analogy to the finite-dimensional case is important and is not highlighted enough in the paper. Some of the results obtained in the paper transform into simple exercises in optimization, when one considers well posed problems in finite dimensions. In my opinion, it would be beneficial to show (or atleast state) those proofs in the paper, followed by a discussion of the technical difficulties which arise when one moves to infinite dimensions.

- I can appreciate nice math, hence my score. However, in the broader scope of the field, if possible, I would be happy for some motivating examples where one would actually want to consider the specific problems introduced in the paper. The authors mention denoising, deblurring, computed tomography and magnetic resonance imaging. As far as I understand none  of those examples is both 'infinite-dimensional' and 'linear'. Can the authors comment on this point?

- Line 53: Presumably one also wants for the noise $\varepsilon$ to be independent of $x$. I think this is also used later on.

- Line 109: After reading Examples 2.3 I do not feel like I understand the problem better. I mean, Example 2.2 is kind of canonical, so it's nice to keep in mind, but I am not sure what purpose Example 2.3 is supposed to achieve.
I can say the same about Example 2.6. However, this is a matter of personal taste and I leave it to the authors discretion.

- Line 123: I feel that Remark 2.5 is very important and actually fundamental to the paper. Why is only a remark and not an actual part of formulating the problem? (Again, this is a matter of personal taste).

- Line 126: I do not understand the comment that $K$ is not identified with $K^*$. In general, those are not isomorphic spaces and cannot be identified. The same comment repeats again, later in the paper.

- Line 129: Shouldn't it be $v \in K$?

- Line 151: Since $v \in K$. I think $\langle Ax, v \rangle_Y$ should be $\langle Ax, \iota(v) \rangle_Y$. It's ok to omit the embedding, when it's clear from the context, but this has to be stated somewhere and it would help the readability if it would be consistent throughout the paper.

- Line 161: "to be at least as smoothing". This sentence is too vague. I feel that saying something concrete would be more helpful for the reader here.

- Line 164: I tried to understand why the main result is constrained to a linear map between Hilbert spaces, rather than, say, Banach spaces. As far as I see, the display beneath Line 164, where the authors expand the error, is the main reason for that. Can the authors comment on that? If this is so, it might be worthwhile to add a small discussion on this point.

- Line 187: "The optimal h^*". As will be shown in Theorem 3.1, there could be more than optimal $h$. The word 'the' is then not appropriate here.

- Line 190: It would be to have all assumptions written together in one place.

- Display beneath Line 194: It's not very common to write "A = B in some subspace". I think that using a restriction sign "|" or just a projection operator would be more familiar.

- Line 232: The quantity in this line is first defined as $\rho_1$ but treated as a different letter in the following display. (Embarrassingly, I am not sure what is the other letter. Perhaps it is a different font of \rho?).

- Display beneath Line 232: The notation $\mathrm{HS}\left(\cdot, \cdot\right)$ was not properly defined. Presumably, it means the space of Hilbert-Schmidt operators. However, later in the Appendix a different notation is also used for this space ($\mathcal{L}_{HS}$), which might create some confusions.

Line 271: I am not sure whether I agree with the statement in this line. The authors show that the strategies are equivalent when $s' \to \infty$, which seems like a degenerate case. Perhaps the authors mean to think about $s$ as being largish and then say that for any 'reasonable' $m$ the two bounds are close? Anyways, I would be happy for a clarification. To me, it seems that the 'unsupervised' method is superior (which is supported by the experiments).

Line 325: "e.g. with deep learning methods". I am not sure I see the connections. Again, would be happy for a clarification.

Appendix A.1: On one hand, I am happy that the authors added this appendix. It supplies some intuition, definitions and examples. On the other hand, I'm not sure what purpose it actually serves since all the necessary material was presented in the main paper and this appendix was never referenced in the main text.

Proposition A.2: As far as I can see, these are just the same assumptions, used before. This brings me back to the suggestion of having one place to gather all assumptions.

Line 552: what do you mean by independence of $x$ and $\mu$? $\mu$ is deterministic.

Line 558: "such functional" -> "such a functional"





**Time Spent Reviewing:**

8

---

> ### Author Response · Authors · 2021-08-09
> **Reply to Official Review of Paper10535 by Reviewer NW21**
>
> We would like to thank the reviewer for the positive feedback and the many useful comments and suggestions. We address all of them below.
>
> - **Analogy to the finite-dimensional case.** We agree, a detailed comparison with the corresponding result of Section 3 in the finite-dimensional case would certainly be very useful and beneficial, but unfortunately would not fit the page limit of the manuscript. We decided only to point out a very precise reference  [17, Theorem 12.1] in line 209. We would like also to point out that the analogy is limited to the result of Section 3, and the generalization bounds of Section 4 are new also in the finite-dimensional case.
> - **Motivating examples.** In the inverse problems (theoretical) literature it is indeed common to consider a continuous formulation of the mentioned problems via integral operators in infinite-dimensional spaces, in which the signals are modeled by functions on a continuous domain. For this purpose, we will more carefully add pointers to relevant references for each reported application, such as the book [Hanke-Engl-Neubauer, Regularization of inverse problems] for denoising and deblurring, [Natterer, The mathematics of Computerized Tomography] for CT and [Epstein, Introduction to the Mathematics of Medical Imaging] for MRI. Those references should also help to clarify the linear nature of such inverse problems.
> - **Line 53.** We will clarify that $x$ and $\varepsilon$ are independent also in the introduction, at the end of line 52.
> - **Line 109.** The purpose of Example 2.3 is to show, in a particular case, the link between the prior on $x$ and the covariance operator $\Sigma_x$ in a classical setting with a smoothing prior. Unfortunately, the space limit does not allow us to expand this discussion. Example 2.6 has the simple aim of showing that, in the case when $\varepsilon$ is a random variable (and not only a process), the rather complicated construction of Remark 2.5 may be avoided.
> - **Line 123.** We completely agree that the content of Remark 2.5 is fundamental and should not be seen as a side comment. However, the current formulation in a remark allows us to refer to it more easily in the rest of the paper, which is why we decided to use this environment.
> - **Line 126.**  Since $K$ is a Hilbert space, thanks to the Riesz representation theorem one may identify $K$ with its dual $K^*$. However, using the Gel'fand triple formalism, it is preferable to avoid this identification. For instance, in the Sobolev class, one usually identifies the dual of $L^2$ with itself, but the dual of $H^1$, which is a proper subspace of $L^2$, is not identified with $H^1$ and is denoted by $H^{-1}$, so that $L^2$ can be identified with a proper subset of $H^{-1}$, yielding the Gel'fand triple $H^1\subset L^2\subset H^{-1}$.
> - **Line 129.** Yes, we will fix the typo in the revision.
> - **Line 151.** Thanks, we will add $\iota(v)$ in the revision.
> - **Line 161.** Unfortunately, it is not easy to say something more concrete in the general case. In the case when $AB$ and $\Sigma_\varepsilon\iota$ are simultaneously diagonalizable, this condition means that the singular values of  $AB$ should go to $0$ at least as fast as the singular values of $\Sigma_\varepsilon\iota$. If in the revised version we have enough space, we will add this explanation to clarify the initial comment.
> - **Line 164.** We are surely interested in the extension to Banach spaces (which would take into account $\ell^1$ optimization, as mentioned in the third paragraph of Section 6). By now, we deeply rely on the Hilbert space structure, not only for the expansion in line 164 but also for the Gel'fand triple construction, the expression of the quadratic functional, and the properties of the covering numbers.
> - **Line 187.** This is true, in the revised version we will rephrase this sentence to clarify that the optimal parameters are not necessarily unique.
> - **Line 190.** We agree that it would be useful to have the assumptions in one single place. However, we decided to state the assumptions along the way, while the various ingredients were introduced, in order to preserve the flow of the paper. If there is enough space in the revised version, we will add a summary of the assumptions at the beginning of Section 3.
> - **Line 194.** This is true, we will change the notation in the revision.
> - **Line 232.** Thanks, this is a typo: we will correct it. We decided to use $\varrho_1$ because $\rho$ was already used to denote the joint probability distribution.
> - **Line 232.** Yes, this denotes the space of Hilbert-Schmidt operators indeed. In the revision, we will make the notation uniform and mention the meaning in line 233.
> - **Line 271.** Yes, your interpretation is correct, and we agree that the statement in line 271 is not completely correct as it is stated. Since it is just a (not precise) rewording of what is mentioned before, we propose to remove the sentence in line 271 in the revision.
> - **Line 325.** We agree that this is not clear: we were referring to the approaches where the penalty term in the regularization functional is given by a (learned) neural network, as in [19,20,24,25,26]. We will cite these references here to clarify the connection.
> - **Appendix A.1.** It is true that all the basic concepts related to random processes were already presented in section 2. However, since random processes may not be familiar to everyone, we decided to add some discussion and details in the appendix. We forgot to mention the appendix in the main body: we will add a reference in line 121: "The reader is referred to [11] and to Appendix A.1...".
> - **Proposition A.2.** Yes, absolutely, these are the same assumptions used before, and are reported here for the reader's convenience. As mentioned above, we did not find it easy to put all the assumptions in the same place.
> - **Line 552.** This is a typo: $x$ and $\varepsilon$ are independent.
> - **Line 558.** Thanks for noticing, we will fix this in the revision.

---

> > ### Comment · Reviewer_NW21 · 2021-08-18
> > **Response to rebuttal**
> >
> > Thank you very much for the clarifications and for addressing all of my comments.
> > I still like the paper and I will keep the score.

---

### Official Review · Reviewer_Q7iT · 2021-07-16

**Rating:** 6
**Confidence:** 2

**Summary:**

This paper studies the learning of the optimal generalized Tikhonov regularizer for linear inverse problems. Optimality is investigated with regards to the mean-squared error.

The authors show that the optimal parameters are independent of the forward operator and additive noise. Next they look into how the optimal parameters can be learned from finite data for the supervised and unsupervised setting. They bound the excess risk for both settings. Both these bounds turn out to be similar. Lastly, the authors verify the theoretical bounds through numerical simulations.


**Limitations And Societal Impact:**

The authors adequately discuss the limitations of their work.


**Main Review:**

Originality:

The results seem novel and I have not seen these kinds of results for the generalized Tikhonov regularizer before. However, the significance of the bounds could be better appreciated if they are compared to similar results in the literature. Similar could be done for the proof strategies. More generally, the related work could be strengthened by discussing relevant theoretical work on learning regularizers.

Quality:

The similarities between the bounds in Theorem 4.1 and Theorem 4.2 are interesting. As the authors mention, the bounds become equivalent when s tends to infinity (line 271). Please can the authors explain in more detail what this means with regards to linear inverse problems? For example, does it connect to the forward operator, the noise, the ill-posedness of the problem, etc?

The numerical simulations nicely support the claims in Section 4. Another important finding is that the optimal regularization parameters are independent of the forward operator and additive noise. Do the authors also have experiments that show the same decay as in Figures 1 and 2 when different (non-identity) forward operators and noise levels are used? This is intriguing because in practice, when solving inverse problems, often the regularization parameters depend on the ill-posedness of the inverse problem.

Clarity:

The numerical simulation procedure is explained well and details are provided. In general the paper is well organized.

What is the level of noise that is used in the experiments?
Is there any reason why the excess risk increases when the discretization increases from 64 to 256 (especially for supervised learning)?

Is $c \in K$ on line 129 a typo?

Significance:

I believe the results and insights in this paper are important and insightful. I also agree with the authors that the community can build upon this work by further investigating unsupervised methods and other regularizations that have more complex forms.

##############################

After author response: Thank you for responding to my comments. After reading all the reviews and author responses, I will keep my rating. I believe that the proposed changes by the authors will provide multiple clarifications and improve the paper.


**Time Spent Reviewing:**

11

---

> ### Author Response · Authors · 2021-08-09
> **Reply to Official Review of Paper10535 by Reviewer Q7iT**
>
> We would like to thank the reviewer for the positive feedback and the useful comments and suggestions. We address all of them below.
>
> **Originality:**
>
> The rate we obtain can be meaningfully compared with what is found in supervised learning: we refer to the recent results [Blanchard-Mücke, FoCM, 2018] and [Lin-Rudi-Rosasco-Cevher, ACHA, 2020] and references therein. We propose to add these references, together with a short discussion, in Section 4. We moreover point out that we are not aware of papers related to learning a regularizer that address the theoretical questions considered in our manuscript such as generalization bounds.
>
> **Quality:**
>
> In the proposed framework,  large values of $s$ mean that the compact embedding $H\hookrightarrow X$ has rapidly decaying singular values, which is satisfied e.g. when $H$ is a space of much smoother functions with respect to $X$ (see lines 250-252): the larger $s$ is, the smoother the functions in $H$ are. In the context of inverse problems, this is directly connected with the regularity we can  *a priori* assume on the solution $x$. Indeed, as specified by assumption b) at line 237, we need to assume that the optimal parameter $h^\star$ is an element of $H$, and similarly for $B^\star$. Nevertheless, this kind of *a priori* smoothness assumption on the unknown solution $x$ is rather common in inverse problems as well as in machine learning. We propose to add a comment after the assumptions in Section 4.1 to discuss what it means to have a large $s$.
> Moreover, we have also run additional numerical tests in the case of ill-posed operators, and in particular when the singular values of the operator $A$ decay polynomially. In the outcome of the experiments, we observe the same decay of the excess risk as in the denoising case, up to different constants. We propose to include the related numerical results and a comment in a new Appendix A.8.
> We agree with the referee that it is intriguing that the regularisation parameter does not depend on the level of ill-posedness of the inverse problem, but only on the prior on $x$ (this happens because optimality is measured by averaging over $x$ and $\varepsilon$, and not with a fixed datum).  This was already well-known in finite dimension (line 209), and our result generalizes this important fact to the infinite-dimensional case.
>
> **Clarity:**
>
> Throughout the simulations, we consider a noise level of $5$\% (namely, the standard deviation of the noise is $5$\% of the peak value of the signal). We will add this information to the paper. The discrepancy between 64 and 256 may be due to numerical instabilities, especially for small $m$. We reserve to discuss it in following work. Finally, $c \in K$ at line 129 is a typo, which should be corrected to $v \in K$.

---

### Official Review · Reviewer_SbHe · 2021-07-16

**Rating:** 6
**Confidence:** 3

**Summary:**

The paper considers solving linear inverse problems of the form $y=Ax+\epsilon$, where $A:X\to X$ denotes a bounded linear operator. The setup is a Bayesian setup with known distribution of source and noise. To solve this problem the paper focuses on using generalized Tikhonov regularization and solving the following optimization: $\min_x  d_Y(Ax,y)+\|\|B^{-1}(x-h)\|\|^2$. Let  $R_{h,B}(x)$  denote the unique solution of the mentioned optimization and define $L(h,B)=E_{x,y}\|\|R_{h,B}(x)-x\|\|^2$. For a given problem, the question is what is the optimal choice of $B$ and $h$ in terms of minimizing $L(h,B)$. The main result of the paper is to provide a full characterization the minimizers of $L(h,B)$ and show that $(\mu,\Sigma_x^{1/2})$ is a global minimizer of $L(h,B)$. (Here, $\mu$ and $\Sigma_x$ denote the mean and covariance of the input, respectively.) The paper also studies the case where instead of the actual source and noise distributions we have access to training data. In that case it studies two different methods for learning $(h,B)$ from data.

**Limitations And Societal Impact:**

Yes.

**Main Review:**

Here are my key reservations about the paper and the scope of its novelty:

1) The main result of the paper is an extension of a known result for finite-dimensional spaces to infinite-dimensional spaces. The authors do not motivate the necessity and relevance of such as extension. The problems they describe in the Introduction are already addressed by the prior art. Moreover, the extension, as far as I can tell, is very technical but straightforward.

2) The title of the paper is misleading. The paper focuses on a very specific type of regularization, namely generalized Tikhonov regularization. Clearly, this type of regularization is not optimal for many structured sources of interest. Constrained to this specific (and non-optimal) regularization, the paper characterizes the optimal parameters of the regularizer. This is far from "learning the optimal regularizer". Moreover, for finite-dimensional data, there are known results in the literature (not cited in the paper) on Bayesian high-dimensional linear regression that in fact characterize the optimal regularizer dependent on the source distribution, at least in the cases of noise-free measurements.



**Time Spent Reviewing:**

2

---

> ### Author Response · Authors · 2021-08-09
> **Reply to Official Review of Paper10535 by Reviewer SbHe**
>
> We are grateful to the reviewer for the insightful comments, which point out important aspects of our paper. In the following, we hope to provide a satisfactory answer to the issues raised by the review.
>
> 1.  Many inverse problems are by their own nature infinite-dimensional. In real applications, their reduction to a finite-dimensional problem is mainly due to computational reasons, and the discretization step always introduces an error. Hence, it is of main interest to provide a genuine theoretical analysis in an infinite-dimensional setting, allowing, as a by-product, to bound the discretization error.  Examples of such frameworks are: to solve partial differential equations using finite element methods, medical tomography where the artifacts and ghost images are explained using integral geometry, to solve stochastic differential equations in mathematical finance by Itô-calculus. Note that the properties of the infinite-dimensional inverse problems have been shown to lead to convergence of the solutions only in a low regularity space [Kekkonen-Lassas-Siltanen, Inverse problems 2016] as the dimension of the approximated model goes to infinity. Other relevant references discussing the importance of modeling inverse problems directly in infinite dimensions are [16,30] and [Monard-Nickl-Paternain, CPAM 2021]. In recent years, much research has been done also on machine learning for functions between infinite-dimensional spaces, see [9,22,31], [Parhi-Nowak, JMLR, 2021]. For example, the work of A. Stuart's group on neural operators is an infinite-dimensional generalization on neural networks displaying genuinely new features, see e.g. [Nelsen-Stuart, arxiv 2020] and [Li et al., ICLR 2021]. According to these considerations, we thus believe that the infinite-dimensional framework considered in our manuscript has solid motivations and we propose to include (part of) this discussion in Section 1.
> We agree with the reviewer that the statements of our results are closely related to their finite-dimensional counter-part, however, the assumptions and the proofs are very different. For example, we need to consider random processes, to introduce Gel'fand triples, to modify the objective function, compare (10) with (11). Furthermore, we need to assume Assumption 2.8, which is new and at the root of our proofs. We also would like to point out that the generalization bounds provided in Section 4 are, to the best of our knowledge, new also in a finite-dimensional context.
> 2. We chose the proposed title trying to balance a rigorous and precise description of our work and the desire to avoid technical expressions. We are nevertheless open to discuss it and modify it. Following the fair criticism of the reviewer, a possible updated title could be *Learning the optimal Tikhonov regularizer for inverse problems*.
> We are moreover happy to add references to results in the literature regarding optimal (non-quadratic) regularizers associated with different priors, e.g. [Gribonval, IEEE Trans. Signal Process. 2011], [Gribonval-Machart, NIPS 2013] and [Burger-Lucka, Inverse Problems 2014], which always deal with a finite-dimensional setting (we would be happy to hear about other references we are not aware of). We plan to cite them in the third paragraph of Section 6 and we plan to add a comment about the relation with our contribution.

---

### Decision · Program_Chairs · 2021-09-27

**Decision:**

Accept (Poster)

**Comment:**

This paper studies a basic problem in optimal estimation, whose answer is well-known in the finite-dimensional case, and extends the characterization to inifinite-dimensional settings. In particular suppose we are given an observation of the form $A x + \epsilon$ and our goal is to estimate $x$ and achieve minimum mean-squared error. This paper studies the problem of characterizing the optimal regularizer, when constrained to a Tikhonov regularization scheme. They show that the optimal regularizer is independent of $A$ (this depends crucially on the assumption that $x$ and $\epsilon$ are independent) and moreover they give finite-sample guarantees for being able to approximate it. While the results are not surprising, there are many subtleties that just don’t happen in the finite dimensional case (e.g. $A$ will not in general have a bounded inverse). Overall the paper studies an important problem, makes a solid contribution and is well-written.